# VoLTA: Vision-Language Transformer with Weakly-Supervised Local-Feature Alignment

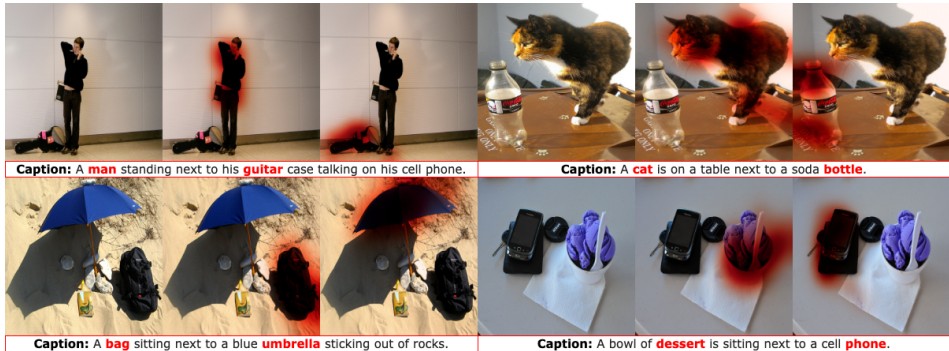

Figure 1: We introduce VoLTA, **Vi**sion-**L**anguage **T**ransformer with weakly-supervised local-feature **A**lignment, a VLP paradigm trained with graph optimal transport (GOT) based *image-text* matching. VoLTA learns fine-grained local visual representation only using global *image-caption* pairs, eliminating the use of expensive grounding annotations. **This figure shows how different words in captions attend relevant image regions, produced by the GOT module of VoLTA pre-trained on COCO.**

## ABSTRACT

Vision-language pre-training (VLP) has recently proven highly effective for various uni- and multi-modal downstream applications. However, most existing end-to-end VLP methods use high-resolution image-text-box data to perform well on fine-grained region-level tasks, such as object detection, segmentation, and referring expression comprehension. Unfortunately, such high-resolution images with accurate bounding box annotations are expensive to collect and use for supervision at scale. In this work, we propose VoLTA (**Vi**sion-**L**anguage **T**ransformer with weakly-supervised local-feature **A**lignment), a new VLP paradigm that only utilizes image-caption data but achieves fine-grained region-level image understanding, eliminating the use of expensive box annotations. VoLTA adopts graph optimal transport-based weakly-supervised alignment on local image patches and text tokens to germinate an *explicit*, *self-normalized*, and *interpretable* low-level matching criterion. In addition, VoLTA pushes multi-modal fusion deep into the uni-modal backbones during pre-training and removes fusion-specific transformer layers, further reducing memory requirements. Extensive experiments on a wide range of vision- and vision-language downstream tasks demonstrate the effectiveness of VoLTA on fine-grained applications without compromising the coarse-grained downstream performance, often outperforming methods using significantly more caption and box annotations.

## 1 INTRODUCTION

Inspired by the escalating unification of transformer-based modeling in vision (Dosovitskiy et al., 2021; Liu et al., 2021; Chen et al., 2021a) and language (Devlin et al., 2019; Liu et al., 2019) domains, coupled with readily available large-scale *image-caption* pair data, vision-language pre-training (VLP) (Lu et al., 2019; Li et al., 2020a; Kim et al., 2021; Kamath et al., 2021; Zhang et al., 2021) has recently been receiving ever-growing attention. VLP has not only been proven the *de-facto* for several VL tasks, but it has also been beneficial for traditional vision-only tasks, such as image classification and object detection. Such wide-range applications of VLP can broadly be categorized into two groups: ($i$) tasks requiring image-level understanding, e.g., image classification, image &

text retrieval (Plummer et al., 2015), visual question answering (Antol et al., 2015), and $(ii)$ tasks requiring region-level understanding, e.g., object detection, instance segmentation, and referring expression comprehension (Kazemzadeh et al., 2014; Yu et al., 2016). Most existing VLP methods support either application, leaving the question of a generalizable and unified VL framework.

Traditional VLP methods with image-level understanding (Li et al., 2021a; Wang et al., 2021b; Dou et al., 2022b) utilize large-scale *image-caption* pair datasets and are commonly trained with image-text contrastive objectives computed on global features. Hence, it is not trivial to extend such methods to region-level applications. On the other hand, VLP methods with region-level understanding (Kamath et al., 2021; Li et al., 2022c; Zhang et al., 2022) use *image-text-box* grounding data and are designed to predict bounding boxes during pre-training. Consequently, they do not support image-level tasks. Furthermore, accurate bounding box annotations require high-resolution input images, which are often expensive to collect, annotate and use for pre-training at scale. Recently, FIBER (Dou et al., 2022a) addressed the problem of such unified VLP and proposed a two-stage pre-training algorithm requiring fewer box annotations than previous region-level pre-training methods. Moving a step forward, we aim to eliminate the use of costly box annotations and ask the challenging but natural question: *Can we attain region-level understanding from global image-caption annotations and unify image- and region-level tasks in a single VL framework?*

Subsequently, we focus on achieving region-level fine-grained understanding by weakly-supervised alignment of image patches and text tokens. Previous VLP methods (Chen et al., 2020d; Kim et al., 2021) in this direction use Wasserstein distance (WD) (Peyré et al., 2019), a.k.a Earth Mover's distance (EMD)-based optimal transport (OT) algorithms for such alignment problem. However, we argue that WD is not optimum for intricate images with multiple similar entities. Thus, we propose to jointly utilize Gromov-Wasserstein distance (GWD) (Peyré et al., 2016) and Wasserstein distance (WD) in a setup known as graph optimal transport (Chen et al., 2020a). Moreover, instead of using commonly deployed contrastive objective, we propose to utilize redundancy reduction from Barlow Twins (Zbontar et al., 2021), which is less data-intensive and does not require hard-negative mining. We also follow Dou et al. (2022a) to incorporate deep multi-modal fusion into the uni-modal backbones, removing the need for costly fusion-specific transformer layers. To this end, we introduce VoLTA, **Vi**si**o**n-**L**anguage **T**ransformer with weakly-supervised local-feature **A**lignment, a unified VLP paradigm only utilizes *image-caption* annotations but achieves fine-grained region-level image understanding, eliminating the use of expensive box annotations. Figure 1 visualizes the feature-level image-text alignment. VoLTA can attend text tokens to the corresponding visual patches without relying on low-level supervision.

In summary, our contributions are three-fold. $(i)$ We propose to use graph optimal transport for weakly-supervised feature-level patch-token alignment. $(ii)$ We introduce VoLTA, a unified VLP paradigm for image-level and region-level applications, but pre-trained only using *image-caption* pairs. VoLTA is memory, compute, and time-efficient and can easily be scaled up with readily available large-scale *image-caption* data harvested from the web. $(iii)$ We performed a wide range of vision- and vision-language coarse- and fine-grained downstream experiments to demonstrate the effectiveness of VoLTA compared to strong baselines pre-trained with significantly more caption and box annotations.

## 2 RELATED WORKS

**Uni-modal Self-supervised Pre-training:** In recent years, the machine learning community has observed a boom of self-supervised pre-training. In the language domain, representations learned by BERT (Devlin et al., 2019), RoBERTa (Liu et al., 2019) have become the default setting for any downstream tasks. Generative models such as GPT (Radford et al., 2019; Brown et al., 2020) have also achieved impressive few-shot/zero-shot performances on novel applications. SimCSE (Gao et al., 2021) uses contrastive learning to help learn useful sentence representations.

In the vision domain, a series of contrastive/joint-embedding methods (He et al., 2020; Chen et al., 2020c; 2021b; 2020b; Grill et al., 2020; Chen & He, 2021; Caron et al., 2021; Zbontar et al., 2021; Bardes et al., 2022; Assran et al., 2022) have outperformed supervised counterparts. Recently, generative models such as BEiT (Bao et al., 2021) and MAE (He et al., 2022) have also achieved impressive performances with much more scalable potential.

**Vision-language pre-training:** Vision-language pre-training (VLP) mainly relies on *image-text* pair datasets to learn joint visual-language representations. One line of work is to train separate vision and

language encoders and only fuse their representation in the representation space. CLIP (Radford et al., 2021), UniCL (Yang et al., 2022a) and ALIGN (Jia et al., 2021) use image-text contrastive loss to learn aligned representations. SLIP (Mu et al., 2021) combines self-supervised visual representation learning and contrastive multi-modal learning. M3AE (Geng et al., 2022), FLAVA (Singh et al., 2022) combines masked image modeling and masked language modeling. Another line of work is to use cross attention to fuse vision and language information in the early stage (Kamath et al., 2021; Dou et al., 2022b; Lu et al., 2019; Li et al., 2020b; Kiela et al., 2019; Kim et al., 2021; Zhang et al., 2021; Li et al., 2022b; Wang et al., 2022c). These works focus on learning semantic-level aligned vision-language representations. In addition, UniTAB (Yang et al., 2022c), OFA (Wang et al., 2022b), GLIP Li et al. (2022c), and FIBER (Dou et al., 2022a) use expensive grounding *image-text-box* annotations to learn fine-grained aligned representations. Our work uses representation space alignment and cross-attention fusion, but we do not use any box annotation to learn robust feature-level alignments.

**Unsupervised Representation Alignment:** Unsupervised multi-modal alignment typically relies on specific metrics. Wasserstein distance (WD) (Peyré et al., 2019), a.k.a Earth Mover's distance (EMD)-based optimal transport (OT) algorithms have widely been adopted to various domain alignment tasks, including sequence-to-sequence learning (Chen et al., 2019), few-shot learning (Zhang et al., 2020), knowledge distillation (Balaji et al., 2019), unsupervised domain adaptation (Balaji et al., 2019), generative networks (Han et al., 2015; Genevay et al., 2018; Mroueh et al., 2018; 2019), and multi-modal learning (Yuan et al., 2020; Chen et al., 2020d; Kim et al., 2021; Li et al., 2022d; Pramanick et al., 2022). Previous VLP methods (Chen et al., 2020d; Kim et al., 2021), which use OT-based patch-word alignment, only utilize WD. However, we argue that jointly modeling Gromov-Wasserstein distance (GWD) (Peyré et al., 2016) and Wasserstein distance (WD) results in a superior multi-modal alignment for intricate images. To the best of our knowledge, this is the first work to apply WD and GWD-based optimal transport for feature-level alignment in VLP.

## 3 METHOD

In this section, we present our proposed framework, VoLTA, which contains three broad modules - $(i)$ intra- and inter-modality redundancy reduction, $(ii)$ weekly-supervised cross-modal alignment (CMA) of local features, and $(iii)$ cross-modal attention fusion (CMAF). Next, we introduce the fine-tuning strategies for various uni- and multi-modal downstream tasks as supported by VoLTA. An overview of different modules of VoLTA is depicted in Figure 2.

### 3.1 PROPOSED SYSTEM - VoLTA

#### 3.1.1 INTRA- AND INTER-MODALITY REDUNDANCY REDUCTION

We use Barlow Twins (BT) (Zbontar et al., 2021), a non-contrastive covariance regularization as the foundational objective of VoLTA. The recent success of contrastive vision-language pre-training (VLP) (Radford et al., 2021; Li et al., 2021b; Jia et al., 2021; Kim et al., 2021; Yang et al., 2022a; Dou et al., 2022a;b) has already shown that, compared to a single modality, *image-caption* pairs offer a significantly higher-level of abstractive and semantic concepts about the training samples. However, common contrastive VLP objectives, like InfoNCE (Oord et al., 2018), are data-hungry, as they require large batch sizes and well-mined hard negatives. On the other hand, the BT objective operates on the dimensions of the embeddings across the two views of training samples. Hence, it has proven robust to batch size and can be trained using lower memory resources. In this work, we extend the BT objective for multi-modal setup.

The original BT algorithm, which operates on joint embeddings of distorted samples, was proposed only for image modality. Specifically, for each image of a batch $\mathcal{X}$, two distorted views are obtained using a distribution of data augmentation $\mathcal{T}$ with disparate probabilities. These distorted images are then fed into a shared image encoder which contains a feature extraction network (e.g., ResNet (He et al., 2016)) cascaded with trainable linear projection layers, producing a batch of parallel embeddings $z^A$ and $z^B$. The BT loss computed using the encoded embeddings can be denoted as:

$$\mathcal{L}_{\text{BT}} \triangleq \sum_i \left(1 - C_{ii}\right)^2 + \lambda \sum_i \sum_{j \neq i} \left(C_{ij}\right)^2, \quad C_{ij} = \frac{\sum_b z_{b,i}^A z_{b,j}^B}{\sqrt{\sum_b \left(z_{b,i}^A\right)^2} \sqrt{\sum_b \left(z_{b,j}^B\right)^2}} \quad (1)$$

Where $\lambda$ is a positive weighting factor; $C$ is the cross-correlation matrix computed between $z^A$ and $z^B$ along the batch dimension; $b$ stands for sample indices in a batch; $i, j$ refers to the dimension

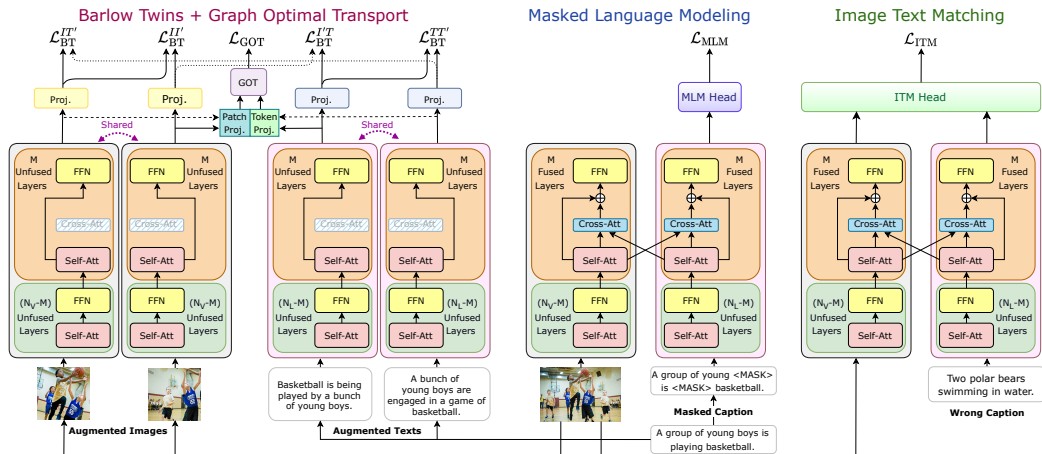

Figure 2: **Computation of four different objectives, $\mathcal{L}_{BT}$, $\mathcal{L}_{GOT}$, $\mathcal{L}_{MLM}$, and $\mathcal{L}_{ITM}$ by the proposed framework, VoLTA** (**Vis**ion-**L**anguage **T**ransformer with weakly-supervised local-feature **A**lignment). Inspired by (Dou et al., 2022a), VoLTA inserts cross-modal attention fusion (CMAF) inside uni-modal backbones with a gating mechanism. During VoLTA pre-training, every forward iteration consists of three steps - $(i)$ CMAF is switched off, VoLTA acts as dual encoder, $\mathcal{L}_{BT}$ and $\mathcal{L}_{GOT}$ are computed. $(ii)$ CMAF is switched on, VoLTA acts as fusion encoder, image-masked caption pair is fed into the model to compute $\mathcal{L}_{MLM}$. $(iii)$ CMAF is kept on, randomly sampled image-caption pair is fed into the model to compute $\mathcal{L}_{ITM}$. Such fusion strategy results in a lightweight and flexible model compared to using fusion-specific transformer layers.

indices of $z^A$ and $z^B$. The first term in Equation 1 is the *invariance* term which attempts to equate the diagonal elements of the cross-correlation $C$ matrix to 1, whereas the second term is the *redundancy reduction term* which pushes the off-diagonal elements of the $C$ matrix to 0.

In this work, we propose to use BT for *image-caption* pairs. Specifically, we use stochastic data augmentations for both images and texts[1], and directly apply the BT objective for all the $2 \times 2$ pairs, resulting in additional supervision. Note that, this simple, straightforward and instinctive extension enables us to apply redundancy reduction in-between and across modalities, which intuitively results in superior visual representation. Moreover, in this bi-modal setting, we can pre-train a text encoder in parallell with the image encoder, and thus, can generalize our system to a wider range of uni- and multi-modal downstream applications.

**Intra-modal Objective:** Intra-modal objective refers to applying the BT loss in-between the pairs of image and text embedddings. Given an *image-caption* pair, we first have two augmented views $(I, I')$ for each image, and two augmented views $(T, T')$ for each text. Then, we resort to Equation 1 individually for the image and text pairs.

$$\mathcal{L}_{BT}^{k} \triangleq \sum_{i} \left(1 - C_{ii}^{k}\right)^{2} + \lambda \sum_{i} \sum_{j \neq i} \left(C_{ij}^{k}\right)^{2}, \, \forall \, k \in \{II', TT'\} \quad (2)$$

**Inter-modal Objective:** Inter-modal objective refers to applying the BT loss across the image and text embedddings. Since the image and text encoder can output features with different shapes, we design the projector layers with same output dimension. Hence, in addition to the original BT loss between $(I, I')$ in Zbontar et al. (2021), we get three more loss terms - $(T, T')$, $(I, T')$, $(I', T)$, leading to $3\times$ diverse and high-quality additional supervision. The inter-modal BT losses can also directly be computed following Equation 1.

$$\mathcal{L}_{BT}^{k} \triangleq \sum_{i} \left(1 - C_{ii}^{k}\right)^{2} + \lambda \sum_{i} \sum_{j \neq i} \left(C_{ij}^{k}\right)^{2}, \, \forall \, k \in \{IT', I'T\} \quad (3)$$

The resulting bi-modal BT loss is $\mathcal{L}_{BT} = \sum_{k} \mathcal{L}_{BT}^{k}, \forall \, k \in \{II', TT', IT', I'T\}$.

---

[1] Image and text augmentations details can be found in Appendix D.1

### 3.1.2 ALIGNMENT OF LOCAL FEATURES

Though the inter-modal redundancy reduction provides high-quality semantic supervision, it is computed on the global image- and text features and, thus, only simulates implicit and non-interpretable multi-modal alignment. However, fine-grained region-level downstream applications like detection, segmentation, and reference expression comprehension require local visual feature descriptors with specific spatial information. To achieve this, most existing top-performing VLP methods, including UniTAB (Yang et al., 2022c), OFA (Wang et al., 2022b), GLIP (Li et al., 2022c), and FIBER (Dou et al., 2022a), use high-resolution image-text-box data for fine-grained pre-training. However, bounding box annotations are expensive to collect and use for supervision. Hence, we seek an alternate weekly-supervised solution for local feature-level alignment using global image-caption annotations.

Recently, Wasserstein distance (WD) (Peyré et al., 2019), a.k.a Earth Mover's distance (EMD)-based optimal transport (OT) algorithms have been used for weakly-supervised patch-word alignment in VLP (Chen et al., 2020d; Kim et al., 2021). Such OT-based learning methods are optimized for distribution matching by minimizing the cost of a transport plan. We pose the patch-word alignment as a more structured graph-matching problem and use the graph optimal transport (GOT) algorithm, which utilizes Gromov-Wasserstein distance (GWD) (Peyré et al., 2016) in conjunction with WD to ensure the preservation of topological information during cross-modal alignment. More specifically, we obtain the patch- and token-level features from the last layers of corresponding visual and textual transformer encoders, and use these encoded local-feature vectors to construct modality specific dynamic graphs - $\mathcal{G}_x(\mathbb{V}_x, \mathcal{E}_x)$ for image patches and $\mathcal{G}_y(\mathbb{V}_y, \mathcal{E}_y)$ for text tokens. Each node in these graphs $i \in \{\mathbb{V}_x, \mathbb{V}_y\}$ is represented by corresponding feature vectors, and intermediate edges $e \in \{\mathbb{E}_x, \mathbb{E}_y\}$ by thresholded cosine similarity.

**Importance of GOT in patch-word alignment:** As mentioned previously, GOT adopts two types of OT distances - WD for node matching and GWD for edge matching. In contrast, previous vision-language pre-training algorithms (Chen et al., 2020d; Kim et al., 2021) using OT for patch-word alignment only considered WD. However, we argue that intricate images with multiple similar objects with different shapes and colors require both WD and GWD for accurate, fine-grained matching. For example, in Figure 3, there are multiple *"men"* present in the image. WD can only match nodes in the graph, and will treat all *"men"* entities as identical and will ignore neighbouring relations like *"in blue shirt"* and *"holding the scissors"*. However, by using proper edge matching with GWD, we can preserve the graph's topological structure. We can correctly identify which *"man"* in the image the sentence is refer-

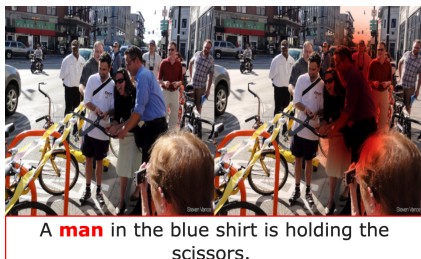

Figure 3: **Example of an intricate image containing multiple similar entities**, and the visual attention map corresponding to the marked word in the caption, produced by the GOT module of VoLTA.

ring to. Hence, we couple WD and GWD mutually beneficially and use a joint transport plan for accurate patch-word matching.

Once $\mathcal{G}_x$ and $\mathcal{G}_y$ are computed, we follow Chen et al. (2020a) to compute WD and GWD.

**Wasserstein Distance (WD)** calculates the pairwise distances between two sets of cross-domain node embeddings. Consider two discrete distributions, $\phi \in \mathbf{P}(\mathbb{X})$ and $\psi \in \mathbf{P}(\mathbb{Y})$, where $\phi = \sum_{i=1}^{n} u_i \delta_{x_i}$ and $\psi = \sum_{j=1}^{m} v_j \delta_{v_j}$; and $\delta_x$ being the Delta-Dirac function centered on $x$. Since $\phi$ and $\psi$ are both probability distributions, sum of weight vectors is 1, $\sum_i u_i = 1 = \sum_j v_j$. The WD distance between $\phi$ and $\psi$ is defined as:

$$\mathcal{D}_{\mathrm{w}}(\phi, \psi) = \min_{\mathbf{T} \in \Pi(u,v)} \sum_i \sum_j \mathbf{T}_{ij} \cdot c(x_i, y_j) \tag{4}$$

where $\Pi(u, v) = \{\mathbf{T} \in \mathbb{R}_+^{n \times m} | \mathbf{T} \mathbf{1}_m = u, \mathbf{T}^\top \mathbf{1}_n = v\}$, $c(x_i, y_j)$ is cosine distance metric, and $\mathbf{T}$ is the transport plan, interpreting the amount of mass shifted from $\phi_i$ to $\psi_j$.

**Gromov-Wasserstein Distance (GWD)** assists in edge matching and preserves graph topology by calculating distances between pairs of nodes in each domain, as well as measuring how these distances compare to the counter domain. In the same discrete graph matching setting, GWD between

$\phi$ and $\psi$ can be mathematically represented as:

$$\mathcal{D}_{\text{gw}}(\phi, \psi) = \min_{\hat{\mathbf{T}} \in \Pi(u,v)} \sum_{i,i',j,j'} \hat{\mathbf{T}}_{ij} \hat{\mathbf{T}}_{i'j'} L(x_i, y_j, x_i', y_j') \tag{5}$$

where intra-graph structural similarity between two node pairs $(x_i, x_i')$ and $(y_j, y_j')$ is represented as $L(x_i, y_j, x_i', y_j') = \|c_1(x_i, x_i') - c_2(y_i, y_i')\|$, $c_i$ being cosine similarity between a node pair in any graph $\mathcal{G}_i$. Transport plan $\hat{\mathbf{T}}$ is periodically updated to align the edges in different graphs belonging to disparate modalities.

We further follow Chen et al. (2020a) to combine WD and GWD transport plans, leading to a unified GOT objective given as:

$$\mathcal{L}_{\text{GOT}}(\phi, \psi) = \gamma \mathcal{D}_{\text{w}}(\phi, \psi) + (1 - \gamma) \mathcal{D}_{\text{gw}}(\phi, \psi) \tag{6}$$

where $\gamma$ regulates the importance of two loss terms.

### 3.1.3 CROSS-MODAL ATTENTION FUSION (CMAF)

BT and GOT losses are computed in a dual encoder setting, which does not contain cross-modal interactions and are not suitable for complex multi-modal feature representation. Most existing methods, including UNITER (Chen et al., 2020d), ViLT (Kim et al., 2021), METER (Dou et al., 2022b) and GLIP (Li et al., 2022c) design cross-modal fusion by stacking additional transformer layers on top of uni-modal encoders, introducing large number of added parameters during pre-training. We follow a more efficient solution proposed by FIBER (Dou et al., 2022a), which inserts cross-modal fusion into the uni-modal backbones with a gating mechanism. Specifically, at the top $M$ transformer layers in the vision and language backbone, cross-attention signals, weighted by a gating scalar $\alpha$, are added to self-attention:

$$\begin{aligned} \hat{x} &= \text{Self-Att}(x) \\ x &= x + \hat{x} + \alpha * \text{Cross-Att}(\hat{x}, y) \\ x &= x + \text{FFN}(x) \end{aligned} \tag{7}$$

where $\alpha$ is a trainable parameter initialized to $0$. Following existing literature (Li et al., 2021a; Wang et al., 2021a; Dou et al., 2022b;a), we use Masked Language Modeling (MLM) and Image-Text Matching (ITM) to pre-train the cross-attention parameters. For MLM, we randomly mask $15\%$ text tokens, and the loss aims to reconstruct the masked tokens. We feed the network with randomly sampled image-caption pairs for ITM, and the loss predicts whether they are matched. The gating mechanism is a good choice for CMAF because $(i)$ cross-attention parameters can easily be switched off by setting the gating scalar $\alpha$ to $0$ when computing the BT and GOT losses. Thus, we can learn the cross-attention parameters without affecting the original computational flow of uni-modal backbones. $(ii)$ gating mechanism is more lightweight and memory-efficient than adding fusion-specific layers (GLIP and METER use $4\times$ more fusion parameters than FIBER).

Overall, VoLTA training pipeline can be summarized in the following three steps:
- **BT & GOT:** CMAF is switched off ($\alpha = 0$), VoLTA acts as dual encoder, $\mathcal{L}_{\text{BT}}$ and $\mathcal{L}_{\text{GOT}}$ are computed.
- **MLM & ITM**: CMAF is switched on ($\alpha \neq 0$), VoLTA now acts as fusion encoder, $\mathcal{L}_{\text{MLM}}$ and $\mathcal{L}_{\text{ITM}}$ losses are computed.
- **Back-propagation:** the four losses are added, giving $\mathcal{L}_{\text{total}} = \mathcal{L}_{\text{BT}} + w_{\text{GOT}} * \mathcal{L}_{\text{GOT}} + \mathcal{L}_{\text{MLM}} + \mathcal{L}_{\text{ITM}}$, and back-propagated into the model end-to-end. An ablation on $w_{\text{GOT}}$ is given in Appendix E.3.

The overall VoLTA pipeline for computation of different training objectives is shown in Figure 2. The pseudo code for VoLTA can be found in Appendix A.

### 3.2 FINETUNING FOR DOWNSTREAM TASKS

We adopt VoLTA to a wide range of vision- and vision-language downstream tasks. We switch off the inserted cross-attention modules for the vision-only tasks and use the image encoder. For the

Table 1: **Uni-modal downstream: linear image classification.** We benchmark learned representations on image classification task by training linear classifiers on fixed features. We report top-1 accuracy on ImageNet-1k validation set, classification mAP on VOC07, and per-class (PC) and overall (O) F1 scores on COCO. Numbers with † are re-implemented by Yuan et al. (2021), and the numbers with ‡ are re-implemented by us. methods trained with significantly larger dataset are colored gray. Best results are in **bold**.

| Method | Pre-train | Arch. | Supervision | Top-1(%) | Method | Pre-train | Arch. | VOC07 SVM | VOC07 MLP | COCO MLP (PC/O) |
|---|---|---|---|---|---|---|---|---|---|---|
| Sup. | IN-1K | RN50 | Label | 76.5 | Sup. | IN-1K | RN50 | 87.5 | 90.8 | 55.2/60.8 |
| Sup. | IN-100 | RN50 | Label | 53.3† | SimCLR | IN-1K | RN50 | 85.5 | – | – |
| MoCo | COCO | RN50 | NA | 44.5† | MoCo | IN-1K | RN50 | 79.8 | – | – |
| MoCo-v2 | COCO | RN50 | NA | 49.3† | MoCo-v2 | IN-1K | RN50 | 86.4 | – | – |
| VirTex | COCO | RN50 | Caption | 52.8 | SwAV | IN-1K | RN50 | 88.9 | – | – |
| ICMLM | COCO | RN50 | Caption | 51.9 | BYOL | IN-1K | RN50 | 86.6 | – | – |
| MCT | COCO | RN50 | Caption | 54.9 | BT | IN-1K | RN50 | 86.2 | 91.9‡ | 56.1/63.0‡ |
| MCT | COCO | RN50 | Caption+Tag | 55.3 | VICReg | IN-1K | RN50 | 86.6 | 91.1‡ | 51.0/57.9‡ |
| VoLTA(w/o CMAF) | COCO | RN50 | Caption | 55.3 | VoLTA(w/o CMAF) | COCO | RN50 | 89.6 | 94.3 | 71.4/74.3 |
| VoLTA(w/o CMAF) | COCO | Swin-T | Caption | 56.3 | VoLTA(w/o CMAF) | COCO | Swin-T | 88.2 | 93.5 | 73.4/75.7 |
| VoLTA(w/o CMAF) | COCO | Swin-B | Caption | 62.5 | VoLTA(w/o CMAF) | COCO | Swin-B | 88.5 | 93.9 | 74.1/76.1 |
| VoLTA | COCO | Swin-B | Caption | 62.5 | VoLTA | COCO | Swin-B | **88.7** | **94.0** | **74.5/76.4** |

Table 2: **Uni-modal downstream: object detection and instance segmentation with fine-tuning.** We benchmark learned representations on VOC07 + 12 object detection task using faster R-CNN (Ren et al., 2015), and on COCO2017 object detection and instance segmentation using mask R-CNN He et al. (2017), both with C4 backbone variant (Wu et al., 2019). Best results are in **bold**.

| Method | Pre-train | Arch. | VOC07+12 det $AP_{all}$ | $AP_{50}$ | $AP_{75}$ | COCO det $AP^{bb}$ | $AP^{bb}_{50}$ | $AP^{bb}_{75}$ | COCO instance seg $AP^{mk}$ | $AP^{mk}_{50}$ | $AP^{mk}_{75}$ |
|---|---|---|---|---|---|---|---|---|---|---|---|
| Sup. | IN-1K | RN50 | 53.5 | 81.3 | 58.8 | 38.2 | 58.2 | 41.2 | 33.3 | 54.7 | 35.2 |
| MoCo-v2 | IN-1K | RN50 | 57.4 | 82.5 | 64.0 | 39.3 | 58.9 | 42.5 | 34.4 | 55.8 | 36.5 |
| SwAV | IN-1K | RN50 | 56.1 | 82.6 | 62.7 | 38.4 | 58.6 | 41.3 | 33.8 | 55.2 | 35.9 |
| SimSiam | IN-1K | RN50 | 57.0 | 82.4 | 63.7 | 39.2 | 59.3 | 42.1 | 34.4 | 56.0 | 36.7 |
| BT | IN-1K | RN50 | 56.8 | 82.6 | 63.4 | 39.2 | 59.0 | 42.5 | 34.3 | 56.0 | 36.5 |
| MoCo | COCO | RN50 | 47.5 | 51.1 | 75.4 | 38.5 | 58.5 | 42.0 | 35.0 | 55.6 | 37.5 |
| MoCo-v2 | COCO | RN50 | 48.4 | 52.1 | 75.5 | 39.8 | 59.6 | 43.1 | 35.8 | 56.9 | 38.8 |
| VirTex | COCO | RN50 | 55.6 | 61.5 | 81.4 | 40.9 | 61.7 | 44.8 | 36.9 | 58.4 | 39.7 |
| MCT | COCO | RN50 | 56.1 | 62.4 | 82.1 | 41.1 | **61.8** | **44.9** | **36.9** | 58.2 | 40.0 |
| VoLTA | COCO | RN50 | **56.6** | **62.7** | **84.4** | **41.9** | **61.8** | 44.8 | 36.5 | **58.5** | **40.8** |
| Sup. | IN-1K | Swin-T | – | – | – | 50.5 | 69.3 | 54.9 | 43.7 | 66.6 | 47.1 |
| MoBY | IN-1K | Swin-T | – | – | – | 50.2 | 68.8 | 54.7 | 43.5 | 66.1 | 46.9 |
| VoLTA | COCO | Swin-T | – | – | – | **50.9** | **69.6** | **55.5** | **43.8** | **66.9** | **47.5** |
| Sup. | IN-1K | Swin-B | – | – | – | 51.9 | 70.9 | 56.5 | 45.0 | 68.4 | 48.7 |
| VoLTA | COCO | Swin-B | – | – | – | **52.1** | **71.3** | **56.6** | **45.2** | **68.5** | **49.0** |

vision-language tasks, following Dou et al. (2022a), we utilize the learned cross-attention parameters as required. For example, VQA and visual reasoning employ all cross-attention modules, whereas captioning requires only image-to-text cross-attention. Again, during IRTR, we switch off all cross-attentions and use VoLTA in a dual encoder setting. We keep all cross-attention parameters during multi-modal object detection and referring expression comprehension and train an object detection head from scratch using the language-aware image features.

## 4 EXPERIMENTS

**Pre-training & downstream datasets:** First, we pre-train VoLTA on the image-caption pairs of COCO2017 (Lin et al., 2014) split which consists 123k images (118k and 5k for training and validation, respectively) with 5 captions for each image. Following Chen et al. (2020d) and Huang et al. (2021), we scale-up our pre-training corpus by appending the VG dataset (Krishna et al., 2017) with COCO2017, together consisting 231k images. We divide our downstream tasks into three categories - $(i)$ **uni-modal tasks** includes image classification on ImageNet (Deng et al., 2009), VOC07 (Everingham et al., 2010), COCO, object detection on VOC07 + 12, COCO and instance segmentation on COCO. $(ii)$ **multi-modal coarse-grained tasks** includes image-level VL tasks - visual question answering on VQAv2 (Antol et al., 2015), visual reasoning on NLVR$^2$ (Suhr et al., 2019), image- and text retrieval on Flicker30k (Plummer et al., 2015) and captioning on COCO. $(iii)$ **multi-modal fine-grained tasks** includes region-level VL tasks - referring expression comprehension (REC) on RefCOCO, RefCOCO+, RefCOCOg (Kazemzadeh et al., 2014; Yu et al., 2016), and language-conditioned object detection on COCO and LVIS (Gupta et al., 2019). We exclude any overlap between our pre-training and downstream validation/test splits. Detailed statistics of all downstream datasets are given in Appendix C.

**Network architectures:** Following FIBER (Dou et al., 2022a), we adopt Swin-Base (Liu et al., 2021) and RoBERTa-Base Liu et al. (2019) as our vision and text encoders, which are initialized with weights from uni-modal pre-training. We collect patch- and token features from last transformer layers, and feed them into local projector network to compute GOT loss. Furthermore, we apply `AvgPool` on patch and token-features, and fed them into global projector network to compute BT

Table 3: **Multi-modal coarse-grained downstreams: visual question answering, visual reasoning, retrieval and captioning.** We only compare with methods pre-trained on comparable amount of dataset. For captioning, 4 metrics are reported - B@4: BLEU@4, M: METEOR, C: CIDEr, S: SPICE. Best results are in **bold**. VoLTA-B denotes Swin-B backbone.

| Method | #Pre-train Images | VQAv2 | | NLVR² | | F30k IRTR | | Method | #Pre-train Images | COCO Captioning | | | |
|---|---|---|---|---|---|---|---|---|---|---|---|---|---|
| | | dev | std | dev | test-P | IR@1 | TR@1 | | | B@4 | M | C | S |
| *Models pre-trained on COCO (123k) and/or VG (108k)* | | | | | | | | *Models fine-tuned without CIDEr optimization* | | | | | |
| SCAN | 108k | – | – | – | – | 48.6 | 67.4 | VL-T5 | 180k | 34.5 | 28.7 | 116.5 | 21.9 |
| SCG | 108k | – | – | – | – | 49.3 | 71.8 | VL-BART | 180k | 35.1 | 28.7 | 116.6 | 21.5 |
| PFAN | 108k | – | – | – | – | 50.4 | 70.0 | VoLTA-B | 231k | 38.2 | 30.7 | 126.6 | 22.5 |
| MaxEnt | 123k | 54.1 | 54.8 | – | – | – | – | VoLTA-GOLD-B | 231k | 38.9 | 30.5 | 128.5 | 23.4 |
| VisualBERT | 123k | 70.8 | 71.0 | 67.4 | 67.0 | – | – | *Models fine-tuned with CIDEr optimization* | | | | | |
| LXMERT | 231k | 72.4 | 72.5 | 74.9 | 74.5 | – | – | VoLTA-B | 123k | 39.4 | 30.3 | 132.5 | 23.3 |
| SOHO | 231k | 73.2 | 73.4 | 76.3 | 77.3 | 72.5 | **86.5** | VoLTA-B | 231k | 39.7 | 30.5 | 133.6 | 23.7 |
| VoLTA-B | 123k | 72.8 | 72.9 | 69.1 | 72.9 | 68.2 | 80.0 | VoLTA-GOLD-B | 123k | 39.8 | 30.5 | **137.5** | 23.6 |
| VoLTA-B | 231k | **74.6** | **74.6** | **76.7** | **78.1** | **72.7** | 83.6 | VoLTA-GOLD-B | 231k | **40.2** | **30.9** | **137.5** | **23.7** |

loss. Both local and global projector networks has 3 linear layers with dimension 2048-2048-1024, with batch normalization and ReLU after first two layers. An ablation on projector dimension is given in Appendix E.4. During downstream tasks, we use the image and text features after the `AvgPool` layer. For CMAF, we insert the cross-attention into the top 6 blocks of the vision and text encoders. Moreover, for direct comparison with existing uni-modal baselines, we re-train VoLTA with ResNet50 (He et al., 2016) and Swin-Tiny image encoders.

**Implementation details:** Detailed implementation setup for pre-training and downstream experiments are given in Appendix D.2 and D.3, respectively.

## 4.1 RESULTS ON VISION-ONLY TASKS

We first experiment on three uni-modal tasks - classification, object detection, and instance segmentation. For a direct comparison with existing ResNet50 and Swin-T baselines, we re-train identical encoders with VoLTA pipeline. Furthermore, since the uni-modal tasks do not utilize cross-attention parameters, we perform an ablation by dropping the CMAF module from VoLTA.

**Image Classification:** Table 1 presents the linear probing results of uni-label classification on ImageNet, and multi-label classification on VOC07 and COCO. For all uni-modal tasks, we report results with COCO pre-training for a fair comparison with existing baselines. For ImageNet, we adopt all COCO baselines from Yuan et al. (2021). Even without the CMAF module, VoLTA achieves better performance than all baselines across three datasets with ResNet50 backbone. The Swin backbones and cross-attention module further improve the performance. For VOC07, we report the results for both SVM and MLP-based linear classifiers. VoLTA with ResNet50 backbone achieves state-of-the-art results on VOC07 SVM evaluation, beating the nearest baseline, SwAV, by 0.7 mAP score. These results indicate the ability of VoLTA to learn effective image-level visual features.

**Object Detection & Instance Segmentation:** Next, we perform two uni-modal region-level tasks - object detection on VOC07 + 12 and COCO2017, and instance segmentation on COCO2017. As shown in Table 2, VoLTA yields the state-of-the-art performance in both tasks across majority metrics. Worth noting, VoLTA, pre-trained with 123k *image-caption* pairs achieves better performance than baselines pre-trained with 1.3M images of ImageNet, proving the efficacy of VLP with fine-grained patch-token alignment over vision-only pre-training.

## 4.2 RESULTS ON COARSE-GRAINED VISION-LANGUAGE TASKS

Next, we perform image-level multi-modal downstream tasks - visual question answering (VQA), visual reasoning, retrieval, and captioning.

**VQA & Visual Reasoning:** As reported in Table 3, VoLTA achieves the best performance on VQA and visual reasoning across the baselines pre-trained with a comparable amount of data. Moreover, on VQA, VoLTA beats LXMERT, which is trained with $2\times$ more data. These results demonstrate the efficacy of our method even when utilizing a mid-scale pre-training corpus.

**Retrieval:** Most existing VLP methods use a fusion encoder for image and text retrieval and feed every image-text pair into the model. Though such fine-tuning often results in higher performance, it introduces quadratic time-cost and is not scalable. Following Dou et al. (2022a), we adopt a more efficient strategy. We drop the cross-attention parameters for this task and compute the dot product of image and text features extracted separately in the dual-encoder setting. As shown in Table 3, even with such an approach, VoLTA produces superior performance among the baselines trained with a similar amount of data, beating all three baselines by a significant margin.

Table 4: **Multi-modal fine-grained downstream: referring expression comprehension.** Methods pre-trained on Im-Txt-Box data or significantly larger amount of Im-Txt data are colored gray. Best comparable results are in **bold**. VoLTA-B denotes Swin-B backbone.

| Method | #Pre-train Data | | RefCOCO | | | RefCOCO+ | | | RefCOCOg | |
|---|---|---|---|---|---|---|---|---|---|---|
| | Im-Txt | Im-Txt-Box | val | testA | testB | val | testA | testB | val | test |
| MAttNet | – | – | 76.4 | 80.4 | 69.3 | 64.9 | 70.3 | 56.0 | 66.7 | 67.0 |
| VLBERT | 3M | – | – | – | – | 71.6 | 77.7 | 61.0 | – | – |
| ViLBERT | 3M | – | – | – | – | 72.3 | 78.5 | 62.6 | – | – |
| Ernie-VL-Large | 4M | – | – | – | – | 75.9 | 82.4 | 66.9 | – | – |
| Rosita | 4M | – | 84.8 | 88.0 | 78.3 | 76.1 | 82.0 | 67.4 | 78.2 | 78.3 |
| UNITER-L | 4M | – | 81.4 | 87.0 | 74.2 | 75.9 | 81.5 | 66.7 | 74.9 | 75.8 |
| VILLA-L | 4M | – | 82.4 | 87.5 | 74.8 | 76.2 | 81.5 | 66.9 | 76.2 | 76.7 |
| *Models pre-trained on Im-Txt-Box data* | | | | | | | | | | |
| MDETR-B | – | 1.3M | 87.5 | 90.4 | 82.7 | 81.1 | 85.5 | 73.0 | 83.4 | 83.3 |
| UniTAB | – | 1.3M | 86.3 | 88.8 | 80.6 | 78.7 | 83.2 | 69.5 | 80.0 | 80.0 |
| X-VLM | 4M | 6.15M | 80.2 | 86.4 | 71.0 | - | - | - | - | - |
| OFA-L | 16M | 3M | 90.1 | 92.9 | 85.3 | 84.5 | 90.1 | 77.8 | 84.5 | 85.2 |
| FIBER-B | 4M | 0.8M | 90.7 | 92.6 | 87.3 | 85.7 | 90.1 | 79.4 | 87.1 | 87.3 |
| VoLTA-B | 123k | – | 83.9 | 86.3 | 79.4 | 73.5 | 78.8 | 65.5 | 75.2 | 75.9 |
| VoLTA-B | 231k | – | **86.1** | **88.6** | **81.8** | **77.0** | **82.7** | **67.8** | **78.3** | **78.3** |

**Captioning:** We perform captioning on the COCO dataset to evaluate if VoLTA can adopt a generation task. We integrate GOLD (Pang & He, 2021) into VoLTA during fine-tuning as it produces significant improvements. As shown in Table 3, our approach maintains superior captioning performance across all baselines pre-trained with comparable data. Using CIDEr optimization further improves performance.

It is worth mentioning that besides achieving a superior result than all baselines using a comparable amount of data on multi-modal coarse-grained tasks, VoLTA also outperforms multiple methods pre-trained using magnitude more data. These results, shown in Table F.1, indicate the effectiveness and generalizability of VoLTA across these tasks.

### 4.3 RESULTS ON FINE-GRAINED VISION-LANGUAGE TASKS

Next, we perform region-level multi-modal downstream tasks - referring expression comprehension (REC) and language-guided object detection.

**REC:** This task aims to localize target objects in an image described by a referring expression phrased in natural language and, thus, perfectly evaluates the fine-grained feature representation capability of VoLTA. As depicted in Table 4, VoLTA beats larger-sized UNITER-L and VILLA-L models on the challenging testB split of both RefCOCO and RefCOCO+. Moreover, VoLTA performs comparably with MDETR and UniTAB, even without being trained on grounding data. These results indicate our model's efficacy in learning fine-grained local visual features.

Table 5: **Multi-modal fine-grained downstream: language-conditioned object detection on COCO and LVIS.** All available baselines are pre-trained on Im-Txt-Box data, and are colored gray. VoLTA-B denotes Swin-B backbone.

| Method | COCO Val 2017 | LVIS MiniVal | | | |
|---|---|---|---|---|---|
| | AP | APr | APc | APf | AP |
| *Models pre-trained on Im-Txt-Box and/or with larger size* | | | | | |
| Mask R-CNN | – | 26.3 | 34.0 | 33.9 | 33.3 |
| MDETR | – | 20.9 | 24.9 | 24.3 | 24.2 |
| GLIP-B | 57.0 | 31.3 | 48.3 | 56.9 | 51.0 |
| GLIP-L | 60.8 | – | – | – | – |
| FIBER-B | 58.4 | 50.0 | 56.9 | 58.1 | 56.9 |
| VoLTA-B | **51.6** | **34.4** | **43.1** | **43.8** | **42.7** |

**Object Detection:** We evaluate VoLTA on two challenging language-conditioned object detection benchmarks - COCO and LVIS. Note that, all existing baselines for this tasks are pre-trained on fine-grained *image-text-box* data, whereas VoLTA only utilizes *image-caption* pairs. As shown in Table 5, VoLTA performs comparatively with these strong baselines. Note that VoLTA beats Mask R-CNN, MDETR, and GLIP-B on LVIS APr, which denotes average precision on rare objects. Thus, we conclude that VoLTA achieves impressive localization ability and robustness, even without utilizing any grounding annotations.

## 5 CONCLUSION

We present VoLTA, a unified VLP paradigm that utilizes image-caption data but achieves fine-grained region-level image understanding, eliminating the use of expensive box annotations. VoLTA adopts graph optimal transport-based weakly supervised patch-token alignment and produces an explicit, self-normalized, and interpretable low-level matching criterion. Extensive experiments demonstrate the effectiveness of VoLTA on a wide range of coarse- and fine-grained tasks. In the future, we plan to pre-train VoLTA on larger-scale datasets.

## REPRODUCIBILITY STATEMENT

The pre-training code of VoLTA can be found in the supplementary material. We also provide a detailed implementation setup for pre-training and downstream experiments in Appendix D.2 and D.3, respectively. After publication, we will provide pre-trained checkpoints and open-source the code on a public repository.

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

# A    PESUDO CODE OF VOLTA

The training psuedo code for VoLTA is as follows:

---

**Algorithm 1** PyTorch-style pseudocode for VoLTA.

---

```python
# f_I: Image Encoder, f_T: Text Encoder
# task_names: string containing task names
# I: Image input, T: Text input, N: Batch size, D: Projector dim
# BT: Barlow Twins loss function
# WD, GWD: Wasserstein and Gromov-Wasserstein loss functions
# MLM, ITM: MLM and ITM loss functions, respectively
# gamma: coefficient of GWD loss in GOT
# w_GOT: weight of GOT loss

def GOT(x_1, x_2, f_1, f_2):
    # compute embeddings
    z_A, z_B = f_1(x_1), f_2(x_2) # N x D

    # normalize representation along batch dimension
    z_A_norm = (z_A - z_A.mean(dim=0)) / z_A.std(dim=0)
    z_B_norm = (z_B - z_B.mean(dim=0)) / z_B.std(dim=0)

    # cosine distance matrix
    c = cosine_dist_matrix(z_A, z_B)}
    # Wasserstein distance
    loss_w = W_D(c, z_A.size(0), z_A.size(1), z_B.size(1))
    # Gromov-Wasserstein distance
    loss_gw = GW_D(z_A.transpose(2,1), z_B.transpose(2,1))

    return gamma * torch.mean(loss_gw) + (1 - gamma) * torch.mean(loss_w)

def VoLTA (I, T):
    total_loss = torch.tensor(0.)
    for x in loader: # load a batch with N samples
        # two augmented versions of I, T
        I1, I2 = augment_image(I); T1, T2 = augment_text(T)

        if "BTGOT" in task_names:
            # BT loss
            intra_loss = BT(I1, I2, f_I) + BT(T1, T2, f_T)}
            inter_loss = BT(I1, T1, f_I, f_T) + BT(I2, T2, f_I, f_T)}
            BT_loss = intra_loss + inter_loss
            total_loss += BT_loss

            # GOT loss
            GOT_loss = GOT(I1, T1, f_I, f_T) + GOT(I2, T2, f_I, f_T)}
            total_loss += w_GOT * GOT_loss

        # cross-attention is enabled
        if "MLM" in task_names:

            # MLM loss
            MLM_loss = MLM(T1, I1, mask_T1, f_I, f_T)
            total_loss += MLM_loss

        if "ITM" in task_names:

            # ITM loss
            ITM_loss =ITM(T1, I1, false_image_1, f_I, f_T)
            total_loss += ITM_loss

    return total_loss
```

---

Table B.1: **Overview of VLP models**. OD: objective detector. Xformer: transformer. Emb.: embedding. MLM/MIM: masked language/image modeling. ITM: image-text matching. WRA: word-region alginment. ITC: image-text contrastive learning. Grnd: Grounding. Cap: Captioning. TP: Token Prediction. CA: Contrastive Alignment, NNS: Nearest Neighbour Supervision, MVS: Multiview Supervision, SL: Sim-siam Loss, MHA: Multi-head attn., LM: Language Modeling, UniVLC: Unified Vision Language Contrastive, VLM: Vision Language Matching, BBP: Bounding Box Prediction, MP: Matching Prediction, FSA: Fine-grained Semantic Alignment, TSA: Token-level Semantic Alignment, AMC: Associative Mapping Classification, CMA: Cross-Modal Alignment, IMC: Intra-Modal Contrastive, LMI: Local Mutual Information Maximization, I-T: Image-Text, I-T-B: Image-Text-Box.

| Model | Venue | Vision Encoder | Text Enc. | Multimodality Fusion | Pre-train I-T | Pre-train I-T-B | Pre-training Objectives |
|---|---|---|---|---|---|---|---|
| ViLBERT | NeurIPS'19 | OD+Xformer | Xformer | Co-attn. | ✓ | | MLM+ITM+MIM |
| LXMERT | EMNLP'19 | OD+Xformer | Xformer | Co-attn. | ✓ | | MLM+ITM+MIM+VQA |
| VisualBERT | ACL'20 | | | | ✓ | | MLM+ITM |
| VL-BERT | ICLR'20 | | | | ✓ | | MLM+MIM |
| UNITER | ECCV'20 | OD | Emb. | Merged-attn. | ✓ | | MLM+ITM+MIM+WRA |
| OSCAR | ECCV'20 | | | | ✓ | | MLM+ITM |
| VinVL | CVPR'21 | | | | ✓ | | MLM+ITM |
| VL-T5 | ICML'21 | | | | | ✓ | MLM+ITM+VQA+Grnd+Cap |
| SOHO | CVPR'21 | CNN | Emb. | Merged-attn. | ✓ | | MLM+ITM+MIM |
| SimVLM | ICLR'22 | CNN | Emb. | Merged-attn. | ✓ | | PrefixLM |
| MDETR | ICCV'21 | | Xformer | | | ✓ | OD+TP+CA |
| ViLT | ICML'21 | Patch Emb. | Emb. | Merged-attn. | ✓ | | MLM+ITM |
| Visual Parsing | NeurIPS'21 | Xformer | | | ✓ | | MLM+ITM+MIM |
| ALBEF | NeurIPS'21 | Xformer | Xformer | Co-attn. | ✓ | | MLM+ITM+ITC |
| METER | CVPR'22 | | Xformer | Co-attn. | ✓ | | MLM+ITM |
| CLIP | ICML'21 | CNN/Xformer | | None | ✓ | | ITC |
| DeCLIP | ICLR'21 | CNN/Xformer | | None | ✓ | | ITC+MLM+SL+MVS+NNS |
| ALIGN | ICML'21 | CNN | | None | ✓ | | ITC |
| GLIP | CVPR'22 | OD+Xformer | | | ✓ | ✓ | OD+CE+WRA |
| GLIPv2 | NeurIPS'22 | OD+Xformer | | Cross-modality MHA | ✓ | ✓ | OD+CE+WRA+MLM |
| BLIP | ICML'22 | | Xformer | Cross-modality MHA | ✓ | | ITC+ITM+LM |
| OmniVL | NeurIPS'22 | | | | ✓ | | UniVLC+VLM+LM |
| X-VLM | ICML'22 | Xformer | | | ✓ | ✓ | BBP+ITC+MP+MLM |
| CMAL | ACM MM'22 | | | | ✓ | | AMC+MLM+MRM+ITM+ITC |
| LOUPE | NeurIPS'22 | | | None | ✓ | | ITC+FSA+TSA |
| FILIP | ICLR'22 | | | None | ✓ | | ITC |
| UniCL | CVPR'22 | CNN/Xformer | | | ✓ | | ITC |
| UniTAB | ECCV'22 | CNN | | Merged Attn | | ✓ | LM |
| TCL | CVPR'22 | Xformer | | Merged Attn | ✓ | | CMA+IMC+LMI+ITM+MLM |
| MS-CLIP | ECCV'22 | Xformer | | Shared Attention | ✓ | | ITC |
| FIBER | NeurIPS'22 | Xformer | Xformer | Merged Co-attn | ✓ | ✓ | MLM+ITM+ITC |
| **VoLTA** | – | CNN/Xformer | Xformer | Merged Co-attn | ✓ | | BT+GOT+MLM+ITM |

# B    OVERVIEW OF VISION-LANGUAGE PRE-TRAINING MODELS

Vision-Language Pretrained (VLP) models have proven extremely beneficial for multi-modal tasks in the recent years. Earlier works were predominantly focused on using pre-trained object detectors to extract patch (region) level information from corresponding images (Lu et al., 2019; Li et al., 2020a; Tan & Bansal, 2019; Chen et al., 2020d; Su et al., 2019). In some of these models, such as ViLBERT (Lu et al., 2019) and LXMERT (Tan & Bansal, 2019), multi-modality fusion has been achieved via co-attention which utilizes a third transformer containing fused information independently obtained from respective vision and language encoders (transformer). On the contrary, VisualBERT (Li et al., 2020a), VL-BERT (Su et al., 2019) and UNITER (Chen et al., 2020d) employ a merged attention strategy to fuse both image patches and text features together into a unified transformer through corresponding image and text embedders. In addition to image patches and texts in a unified transformer, OSCAR (Li et al., 2020b) uses object tags as inputs. VinVL (Zhang et al., 2021) follows a similar strategy to that of OSCAR, the only difference being their novel 3-way contrastive loss which optimizes the training objectives used for VQA and text-image matching. VL-T5 (Cho et al., 2021) exploits bounding-box coordinate information, image IDs, and region IDs along with ROI features for visual embedding. Encoded visual and textual features are fed into a bi-directional multi-modal encoder and an auto-regressive text decoder framework for pre-training.

In all the aforementioned methods, pre-trained object detectors are usually frozen during the training, and extracting region-level features from images can be tedious. To address these shortcomings, end-to-end pre-training methods have been developed. PixelBERT (Huang et al., 2020) uses convolutional neural networks (CNNs) based visual encoder and sentence encoder to obtain image and text representations, respectively. These representations are then fed to a subsequent transformer via a cross-modality alignment. SOHO (Huang et al., 2021) uses grid features discretization via a learned

vision dictionary which are then fed into a cross-modal module. SimVLM (Wang et al., 2021b) uses CNN and text token embedding for image and text feature representation extraction along with a unified encoder-decoder transformer trained on a PrefixLM objective. Finally, MDETR (Kamath et al., 2021) uses CNN and RoBERTa (along with corresponding projection layers) for image and text feature extraction, which are then concatenated before passing through a unified transformer trained on 1.3M Image-Text-Box (I-T-B) annotated data.

In recent years, the rise of Vision Transformers (ViT) (Dosovitskiy et al., 2021) has motivated the research community to have an all-transformer framework by incorporating ViTs (instead of CNN backbones) in VLP models. Image patch features and text token embeddings are fed directly into a ViT model for pre-training in VILT (Kim et al., 2021). Visual Parsing (Xue et al., 2021), ALBEF (Li et al., 2021a) and METER (Dou et al., 2022b) use ViTs as vision encoders for image feature generation. For multimodal fusion, ALBEF and METER use co-attention in their pre-training frameworks.

Another class of VLP models in the form of CLIP (Radford et al., 2021), DeCLIP (Li et al., 2021b) and ALIGN (Jia et al., 2021) have been introduced lately. Although, known for their impressive zero-shot recognition ability and excellent transferability to downstream tasks, these models typically rely on huge amount of image-text pairs for pre-training. Contrastive loss forms the core component of the pre-training objectives in these VLP models. In such models (e.g., CLIP (Radford et al., 2021), DeCLIP (Li et al., 2021b)), separate encoders have been used for each modality. On the contrary, Modality-Shared Contrastive Language-Image Pre-training (MS-CLIP) (You et al., 2022) leveraged knowledge distribution across multiple modalities (image and text) through parameter sharing. In their unified framework, the parameters which are being shared between two modalities include the attention and feedforward modules, and the LayerNorm layers.

GLIP (Li et al., 2022c) and GLIPv2 (Zhang et al., 2022) use a localization loss along with a word-region alignment loss for pre-training corresponding encoders using image-text-box annotations. BLIP (Li et al., 2022b) employs image and text encoders connected through a cross-modality multi-head attention which are pre-trained on image-text pairs using contrastive and language modeling objectives. OmniVL (Wang et al., 2022a) utilizes a unified image(and video) encoder and a text encoder pre-trained on image-text, image-label, video-text and video-label pairs using unified vision-language contrastive, vision-language matching and language modeling losses. Furthermore, a visual-grounded alignment decoder is also present for facilitating better learning and alignment between various modalities. X-VLM (Zeng et al., 2022) employs vision transformer to extract features from the subset of patches representing images/regions/objects. These patch features are then paired with associated text features for contrastive learning, matching, and MLM. Additionally, image and text pairings are also done for bounding-box prediction which is used to locate visual concepts in the image. CMAL (Ma et al., 2022) proposes interactions between features (obtained from respective image and text encoders) via cross-modal associative mappings which help in fine-grained semantic alignment between the learned representations. LOUPE (Li et al., 2022a) implements token-level and semantics-level Shapley interaction modeling with global image-text contrastive loss (in a dual-encoder setting) for explicit learning of fine-grained semantic alignment between visual regions and textual phrases without using expensive bounding-box annotations. FILIP (Yao et al., 2022) removes the need for cross-modality attention fusion by modeling the fine-grained semantic alignment between visual and textual tokens via a novel cross-modal late interaction mechanism in the contrastive loss. TCL (Yang et al., 2022b) uses global cross-modal alignment, intra-modal alignment and local mutual information maximization losses along with Masked Language Modeling and Image-Text Matching to learn robust image-text representations during pretraining. UniCL (Yang et al., 2022a) utilizes a unified learning method with a two-way contrastive loss (image-to-text and text-to-image) in the image-text-label space which can learn representations from either of the image-label and image-text data or both. UniTAB (Yang et al., 2022c) employs a transformer based encoder-decoder framework which can jointly output open-ended text and box, encouraging alignment between words and boxes.

FIBER (Dou et al., 2022a) fuses vision and language encoder backbones through merged co-attention which are then pre-trained on 4M data with two stage pre-training (coarse- and fine-grained). Image-text pairs are used in the coarse-grained pre-training stage which is then followed by a fine-grained pre-training stage with image-text-box annotations. However, these bounding box annotations come with extra overheads. Therefore, in our model, VoLTA, we propose an alternate solution for optimal-transport based local feature-level alignment using global image-caption annotations which performs well not only on coarse-grained tasks (such as VQA and Image Captioning), but also on fine-grained

Table C.1: **Dataset statistics for uni-modal and multi-modal downstream tasks.**

| Modality | Task | Dataset | Image Src | #Images | #Text | Metric |
|---|---|---|---|---|---|---|
| Uni-modal | Image Classfn. | IN−1K | IN−1K | 1.3M | - | Accuracy |
| | | COCO | COCO | 123K | - | F1 |
| | | VOC07+12 | VOC07+12 | 16K | - | mAP |
| | Object Det. | COCO | COCO | 123K | - | $AP^{bb}$ |
| | | VOC07+12 | VOC07+12 | 16K | - | |
| | Instance Seg. | COCO | COCO | 123K | - | $AP^{mk}$ |
| Multi-modal coarse-grained | VQA | VQA | COCO | 204K | 1.1M | VQA-Score |
| | NLVR$^2$ | NLVR$^2$ | Web Crawled | 214K | 107K | Accuracy |
| | IR-TR | Flickr30K | Flickr30K | 32K | 160K | Recall@1 |
| | Captioning | COCO | COCO | 123K | 615K | B@4,M,C,S |
| Multi-modal fine-grained | Ref. Exp. Comp. | RefCOCO | COCO | 20K | 142K | Accuracy |
| | | RefCOCO+ | | 20K | 142K | |
| | | RefCOCOg | | 26K | 95K | |
| | Mul. Obj. Det. | COCO | COCO | 123K | 615K | AP |
| | | LVIS Mini | COCO | 123K | 615K | |

tasks (such as Referring Expression Comprehension and Object Detection). Table B.1 encapsulates an overview of the details of all these aforementioned methods.

## C  DOWNSTREAM DATASETS

Our downstream tasks can be categorized into three groups: uni-modal, multi-modal coarse-grained, and multi-modal fine-grained.

**Uni-modal:** For uni-modal tasks, we fine-tune (and validate) our pre-trained model on ImageNet-1k (Deng et al., 2009) for Image Classification, VOC07+12 (Everingham et al., 2010) for image classification and object detection, and COCO (Lin et al., 2014) for image classification, object detection and instance segmentation.

**Multi-modal Coarse-grained:** Here, we fine-tune (and validate) our pre-trained model on VQAv2 (Antol et al., 2015) for visual question answering, NLVR$^2$ (Suhr et al., 2019) for visual reasoning, Flickr30k (Plummer et al., 2015) for image and text retrieval, and COCO (Lin et al., 2014) for image captioning.

**Multi-modal Fine-grained:** For these tasks, we fine-tune (and validate) our pre-trained model on RefCOCO, RefCOCO+, and RefCOCOg (Kazemzadeh et al., 2014; Yu et al., 2016) for referring expression comprehension, and COCO (Lin et al., 2014) and LVIS Mini (Gupta et al., 2019) for language-conditioned object detection.

It is to be noted that several multi-modal downstream tasks are built based on the COCO dataset where the validation and test splits of these tasks are scattered across the raw COCO splits. Therefore, when pre-training our model, we carefully selected the portion of the COCO dataset which does not overlap with the validation/test splits of these multi-modal downstream tasks.

## D  IMPLEMENTATION DETAILS & HYPER-PARAMETER VALUES

### D.1  DATA AUGMENTATION

We use ResNet50/Swin-T/Swin-B (He et al., 2016; Liu et al., 2021) as image encoder and RoBERTa (Liu et al., 2019) as text encoder. Encoders are followed by a corresponding projector network which is a 3-layer MLP with the configuration [$d$-2048-2048-1024] where $d$ represents the embedding dimension of the encoder's output.

**Image Augmentations:** Two sets of random transformations sampled from an augmentation pool are applied on each input image to generate two disparate distorted views. The augmentation policy is composed of `RandomResizedCrop`, `RandomHorizontalFlip`, `ColorJitter`, `RandomGrayscale`, `GaussianBlur`, and `Solarization` augmentations where `RandomResizedCrop` is applied with a probability of 1.0, whilst the remaining ones are applied randomly with varying probabilities following Zbontar et al. (2021) as outlined in Table D.1.

Table D.1: **Image and text augmentation details.**

| Data Type | View # | Augmentation | Probability |
|---|---|---|---|
| Image | 1 | RandomResizedCrop | 1.0 |
| | 1 | RandomHorizontalFlip | 0.5 |
| | 1 | ColorJitter | 0.8 |
| | 1 | RandomGrayscale | 0.2 |
| | 1 | GaussianBlur | 1.0 |
| | 1 | Solarization | 0.0 |
| | 2 | RandomResizedCrop | 1.0 |
| | 2 | RandomHorizontalFlip | 0.5 |
| | 2 | ColorJitter | 0.8 |
| | 2 | RandomGrayscale | 0.2 |
| | 2 | GaussianBlur | 0.1 |
| | 2 | Solarization | 0.2 |
| Text | 1 | Synonym Replacement | 0.1 |
| | 1 | Random Insertion | 0.1 |
| | 1 | Random Swap | 0.1 |
| | 1 | Random Deletion | 0.1 |
| | 2 | Synonym Replacement | 0.1 |
| | 2 | Random Insertion | 0.2 |
| | 2 | Random Swap | 0.1 |
| | 2 | Random Deletion | 0.2 |

**Text Augmentations:** Two sets of random transformations are applied on input text using EDA (Wei & Zou, 2019) including synonym replacement, random insertion, random swap, and random deletion with different probabilities as outlined in Table D.1.

## D.2 PRE-TRAINING SETUP

Table D.2 shows the details of hyper-parameters used during training.

Our proposed VLP model, VoLTA comprises a vision encoder and a language encoder with a merged co-attention for cross-modality fusion. While conducting experiments, we have considered two types of vision encoder backbones - ResNet-50 (He et al., 2016) and Swin Transformer (Liu et al., 2021). It is to be noted that for fair comparisons with related works (Dou et al., 2022b;a), the input image resolution for ResNet-50 encoder backbone is kept as $224 \times 224$, whereas for Swin-B, it is $384 \times 384$. The output embedding dimension of the image encoder in both cases is 1024. Similarly, to be consistent with Dou et al. (2022b;a), we have selected RoBERTa as the language encoder with vocabulary size of 50265, tokenizer as 'roberta-base', maximum input text length of 30, and an output embedding dimension of 768 (please refer to Table D.2 for more details).

Vision and language encoders are individually followed by projector heads, each consisting of 3 linear layers each with 2048 output units (except for the last one which has 1024 output units) accompanied with a Batch Normalization layer and ReLU activation (for exact configuration please refer to Table D.2). The final projected output denotes the feature representation of the input (image and/or text) which are used for downstream tasks. In order to learn these representations, the embeddings (i.e., output from respective encoders) are fed to the loss function of VoLTA.

The loss function of VoLTA includes four different loss components, namely, multi-modal Barlow Twins for intra- and inter-modality redundancy reduction, GOT for alignment of local features, MLM and ITM together for encouraging cross-modal attention fusion. For MLM, we randomly mask 15%[2] (MLM probability in Table D.2) of the input tokens and the model is trained to reconstruct the original tokens. For ITM, the model predicts whether a given image-text pair is matched.

For optimization, we follow the same protocol as described in Zbontar et al. (2021) where we use the LARS (You et al., 2017) optimizer to train our model for 20 epochs with a batch size of 256. A base LR of 0.1 is used for the weights and 0.0048 for the biases and batch normalization parameters which are then multiplied by a factor of 2. We employ a learning rate warm-up (linear) upto a period of 2 epochs followed by a cosine decay schedule to reduce the LR by a factor of 1000. A weight decay parameter of 1e-6 is used excluding the biases and batch normalization parameters. We conducted

---

[2] Following BERT, we decompose this 15% into 10% random words, 10% unchanged, and 80% with a special token [MASK].

grid search for the GOT loss hyperparameter ($w_{\text{GOT}}$) and we empirically found the best value to be 100.

Table D.2: **Pre-training hyper-parameter details for VoLTA.**

| Hyper-parameters | Notation | Value |
|---|---|---|
| Model | | |
| Img. proj. layer config. | - | $[d_i - 2048 - 2048 - 1024]$ |
| Img. embed. dim | $d_i$ | 1024 |
| Img. reso. (RN-50 & Swin-T) | - | $224 \times 224$ |
| Img. reso. (Swin-B) | - | $384 \times 384$ |
| Txt. proj. layer config. | - | $[d_t - 2048 - 2048 - 1024]$ |
| Txt. embed. dim | $d_t$ | 768 |
| Tokenizer | - | 'roberta-base' |
| Vocab size | - | 50265 |
| MLM prob. | - | 0.15 |
| Max. length of text | - | 30 |
| # Heads of Xformer | - | 12 |
| # Layers of Xformer | - | 12 |
| # Fusion block | - | 6 |
| Dropout rate | - | 0.1 |
| Task names | - | 'BTGOT, MLM, ITM' |
| Training | | |
| Batch size | - | 256 |
| Epochs | - | 20 |
| Lambda_BT | $\lambda$ | 0.005 |
| WD and GWD loss weight | $\gamma$ | 0.1 |
| GOT loss weight | $w_{GOT}$ | 100.0 |
| Optimizer | - | LARS |
| Base LR for weights | - | 0.1 |
| Base LR for biases | - | 0.0048 |
| Momentum | - | 0.9 |
| LR scheduler | - | Cosine LR decay (with linear warm-up) |
| Warm-up steps | - | $2 \times$ Epochs |
| Weight decay | - | 1e-6 |
| End LR factor | - | 0.001 |
| Cosine LR amplitude factor | - | 0.5 |

**Pre-training cost:** Our Swin-B backbone takes 6 hours per epoch to train on 64 V100 GPUs, with per GPU batch-size 4.

## D.3 DOWNSTREAM SETUP

### D.3.1 UNI-MODAL DOWNSTREAM TASKS

**Linear Evaluation:** For ImageNet, the linear classifier has been trained for 100 epochs with batch size of 256, LR of 0.3 and a cosine LR schedule. Cross-entropy loss is minimized with SGDM optimizer (momentum of 0.9) and a weight decay of 1e-6. For both COCO and VOC, the linear classifier has been trained for 100 epochs with AdamW optimizer with batch size of 256, LR of 5e-2 and a weight decay of 1e-6.

**Object Detection:** For training the detection model, the detectron2 library (Wu et al., 2019) has been used. The backbone networks for Faster R-CNN (Ren et al., 2015) and Mask R-CNN (He et al., 2017) has been initialized using our pre-trained model.

For VOC07+12, we used the `trainval` set comprising of 16K images for training a Faster R-CNN (Ren et al., 2015) C-4 backbone for 24K iterations using a batch size of 16 across 8 GPUs using SyncBatchNorm. The initial learning rate for the model is 0.15 which is reduced by a factor of 10 after 18K and 22K iterations. Linear warmup (Goyal et al., 2017) is used with a slope of 0.333 for 1000 iterations.

For COCO, Mask R-CNN (He et al., 2017) with a C-4 backbone on the COCO 2017 train split is used for training and the results are reported on the val split. A learning rate of 0.03 is used and the other parameters are kept the same as in the $1\times$ schedule in detectron2 (Wu et al., 2019).

Table E.1: **Ablation study on different losses of the training objective of VoLTA for multi-modal coarse-grained downstream tasks.** Each model is pre-trained on 231k samples from COCO2017 and VG.

| VoLTA | | | | | VQAv2 | | NLVR² | | F30k IRTR | |
|---|---|---|---|---|---|---|---|---|---|---|
| $\mathcal{L}_{BT}$ | $\mathcal{L}_{GOT}$ $\mathcal{L}_{w}$ | $\mathcal{L}_{gw}$ | $\mathcal{L}_{MLM}$ | $\mathcal{L}_{ITM}$ | dev | std | dev | test-P | IR@1 | TR@1 |
| ✓ | – | – | – | – | 71.5 | – | 72.2 | 72.9 | 70.5 | 81.5 |
| ✓ | – | – | ✓ | ✓ | 72.3 | – | 74.2 | 75.5 | 71.0 | 82.6 |
| ✓ | ✓ | – | ✓ | ✓ | 73.3 | – | 75.2 | 77.0 | 72.4 | 82.8 |
| ✓ | ✓ | ✓ | ✓ | ✓ | **74.6** | **74.6** | **76.7** | **78.1** | **72.7** | **83.6** |

### D.3.2 COARSE-GRAINED MULTI-MODAL DOWNSTREAM TASKS

**Vision-Language Classification (VQAv2 and NLVR²):** Vision-Language Classification task encompasses VQAv2 and NLVR² whose hyper-parameter setup has been taken from METER (Dou et al., 2022b) and FIBER (Dou et al., 2022a). Model finetuning is done with peak learning rates of 2e-5 for the backbones, 1e-4 for the cross-modal parameters, and 1e-3 for the head layer for 10 epochs with a batch size of 512. The image resolutions are set to 576 for VQAv2 and 384 for NLVR² and the models are evaluated with the VQA-Scores for VQAv2 and accuracy for NLVR² (Table C.1).

**Image-Text Retrieval (IRTR):** We follow Dou et al. (2022a) for IR-TR setup for the Flickr30k dataset where the cross-attention layers in the backbones are removed during IR-TR fine-tuning and evaluation. The peak learning rates are set to 2e-5 for the backbones and 1e-4 for the head layer, a batch size of 1024 is considered and each image resolution is set to 576. We evaluate on the Recall@1 metric for both text and image retrieval tasks as outlined in the Table C.1.

**Image Captioning:** For image captioning, only the image-to-text attentions are kept for the cross-modality attention fusion and the model is converted into a standard seq2seq model (Dou et al., 2022a). We used causal mask in the decoding side and the outputs are predicted auto-regressively (Dou et al., 2022a). Models are trained with the cross-entropy loss for 5 epochs with the peak learning rates of 5e-5 for the backbones, and 2.5e-4 for the rest of the parameters followed by a two-stage finetuning. In the first stage, finetuning with GOLD (Pang & He, 2021) is done for 5 epochs with a peak learning rate of 1e-5 for the backbones, since it is efficient and has been proven to be effective when the model input can correspond to different outputs. Second stage finetuning involves CIDEr optimization where the learning rate is further reduced to 1e-6 and the model is trained for 3 epochs. A batch size of 512 is considered in both these cases and a beam size of 5 is used during inference. Evaluation metrics include BLEU (Papineni et al., 2002), METEOR (Banerjee & Lavie, 2005), CIDEr (Vedantam et al., 2015), and SPICE (Anderson et al., 2016) scores as shown in Table C.1.

### D.3.3 FINE-GRAINED MULTI-MODAL DOWNSTREAM TASKS

**Referring Expression Comprehension (REC):** We follow Dou et al. (2022a) for training and evaluation on 3 different datasets (RefCOCO, RefCOCO+, and RefCOCO) where the models are finetuned with a batch size of 16 for 20 epochs. A warmup of 2000 steps with a peak LR of 1e-5 for both the OD head as well as the rest of the model's parameters are used. LR drops twice, once at 67% and the other at 89% of the total number of steps. Horizontal flip augmentation has been turned off during REC training because it was observed in Dou et al. (2022a) that horizontal flip adversely affected the performance, particularly on the RefCOCO dataset. Accuracy is used as the evaluation metric in this case (Table C.1).

**Object Detection:** We follow the training and evaluation setup of Dou et al. (2022a) for text-conditioned (multi-modal) object detection. For both COCO and LVIS datasets, model has been finetuned for 24 epochs with batch size of 32, a LR of 1e-5, and two learning rate drops, once at 67% and the other at 89% of the total number of steps. AP scores are used in this case for model evaluation (Table C.1).

## E ABLATION

We have conducted ablation studies, particularly, on pre-training objectives, GOT loss weight and the dimension of projectors which are summarized below:

Table E.2: **Ablation study on different losses of the training objective of VoLTA for referring expression comprehension tasks.** Each model is pre-trained on 231k samples from COCO2017 and VG.

| VoLTA | | | | | RefCOCO | | | RefCOCO+ | | | RefCOCOg | |
|---|---|---|---|---|---|---|---|---|---|---|---|---|
| $\mathcal{L}_{\text{BT}}$ | $\mathcal{L}_{\text{GOT}}$ $\mathcal{L}_{\text{w}}$ | $\mathcal{L}_{\text{gw}}$ | $\mathcal{L}_{\text{MLM}}$ | $\mathcal{L}_{\text{ITM}}$ | val | testA | testB | val | testA | testB | val | test |
| ✓ | − − | | − | − | 81.7 | 84.1 | 77.8 | 71.2 | 76.6 | 62.2 | 71.7 | 71.7 |
| ✓ | − − | ✓ | ✓ | | 82.7 | 85.2 | 78.1 | 72.0 | 77.7 | 62.5 | 72.8 | 72.7 |
| ✓ | ✓ − | ✓ | ✓ | | 83.9 | 86.6 | 80.5 | 73.9 | 79.5 | 64.1 | 74.6 | 74.3 |
| ✓ | ✓ ✓ | ✓ | ✓ | | **86.1** | **88.6** | **81.8** | **77.0** | **82.7** | **67.8** | **78.3** | **78.3** |

Table E.3: **Ablation study on Intra- and Inter-modal Barlow Twins objective for multi-label image classification on VOC**07 **and COCO.** We report classification mAP on VOC07, and per-class (PC) and overall (O) F1 scores on COCO. Each model is pre-trained on 123k train-val samples from COCO2017.

| $\mathcal{L}_{\text{BT}}^{II'}$ | $\mathcal{L}_{\text{BT}}^{TT'}$ | $\mathcal{L}_{\text{BT}}^{IT'}$ | $\mathcal{L}_{\text{BT}}^{I'T}$ | VOC07 MLP | COCO MLP (PC/O) |
|---|---|---|---|---|---|
| ✓ | − | − | − | 90.8 | 72.2/71.7 |
| ✓ | ✓ | − | − | 91.8 | 74.0/73.7 |
| ✓ | ✓ | ✓ | ✓ | **94.0** | **74.5/76.4** |

## E.1 Ablation on Pre-training objective

We analyze the effectiveness of different pre-training objectives through an ablation on coarse- and fine-grained downstream tasks (Tables E.1 and E.2). First, we pre-train VoLTA only with the multi-modal BT loss. In this setup, VoLTA only acts as a dual encoder; thus, the cross-attention parameters are not pre-trained. Next, we add MLM and ITM loss which helps the model to learn cross-modal information via attention fusion. Next, we add the GOT pre-training objective. Note that GOT adopts two types of OT distances − WD for node matching and GWD for edge matching. As shown in table E.2, the GWD helps to improve the performance of reference expression comprehension across RefCOCO, RefCOCO+, and RefCOCOg datasets. Specifically on RefCOCOg, adding $\mathcal{L}_{\text{gw}}$ yields a significant $4.0\%$ boost in the challenging test set. Since this dataset contains intricate images with multiple similar objects with different shapes and colors, GWD is crucial in distinguishing between them. Overall, this set of experiments demonstrates that all of the objectives are necessary for our model to obtain good performance on different coarse- and fine-grained multi-modal tasks.

## E.2 Ablation on Intra- and Inter-modal Barlow Twins Loss

We verify the effectiveness of the multi-modal Barlow Twins (BT) objective by ablating the intra- and inter-modal terms. The first row of Table E.3 is identical to the original image-only BT objective. Next, we introduce the text branch and add the same BT objective between the two views of the caption. Afterward, we add the inter-modal BT objectives. As shown in Table E.3, each loss term improved the image classification performance, demonstrating the importance of both intra- and inter-modal objectives.

## E.3 Ablation on GOT loss weight

In our loss formulation, we introduce a GOT loss weight $w_{\text{GOT}}$ which regulates the alignment of local features through GOT loss. By conducting a grid-search on uni-modal downstream classification tasks, we assessed the impact of $w_{\text{GOT}}$ as shown in the Table E.4 and experimentally found its best value to be 100 in our case. It is to be noted that a very high value of $w_{\text{GOT}}$ considerably degrades the performance on downstream tasks.

## E.4 Ablation on projector dimension

The design of projector head plays a pivotal role in the downstream performance of the model (Garrido et al., 2022). To investigate the impact of hidden and feature (projector output) dimensions, we have tested 4 different configurations on uni-modal downstream classification tasks. It can be noticed from Table E.5 that an increase in the number of parameters in the projector head does not necessarily lead

Table E.4: **Ablation study on the value of** $w_{\text{GOT}}$**, the weight of GOT loss in** $\mathcal{L}_{\text{total}}$ **in the objective of VoLTA for multi-label image classification on VOC07 and COCO.** We report classification mAP on VOC07, and per-class (PC) and overall (O) F1 scores on COCO. Each model is pre-trained on 123k train-val samples from COCO2017.

| $w_{\text{GOT}}$ | **VOC**07 | **COCO** |
|---|---|---|
| | MLP | MLP (PC/O) |
| 50 | 93.4 | 74.1/76.0 |
| 100 | **94.0** | **74.5/76.4** |
| 200 | 93.2 | 73.1/75.5 |
| 500 | 93.1 | 72.8/75.3 |

Table E.5: **Ablation study on the dimension of local and global projector networks of VoLTA for multi-label image classification on VOC07 and COCO.** We report classification mAP on VOC07, and per-class (PC) and overall (O) F1 scores on COCO. Each model is pre-trained on 123k train-val samples from COCO2017.

| Projector Config. | **VOC**07 | **COCO** |
|---|---|---|
| | MLP | MLP (PC/O) |
| 8192-8192-128 | 91.3 | 71.3/73.0 |
| 8192-8192-256 | 91.9 | 72.2/73.3 |
| 2048-2048-512 | 93.4 | 73.9/76.1 |
| 2048-2048-1024 | **94.0** | **74.5/76.4** |

to an increase in performance. For example, a projector configuration of 8192-8192-256 has roughly 8 times more parameters than 2048-2048-1024, although, the latter performs better in downstream tasks (Table E.5), indicating that the output (feature) dimension of projector plays a crucial role in final performance of the model.

# F    RESULTS ON COARSE-GRAINED VISION LANGUAGE TASKS: COMPARISON WITH METHODS USING MORE PRE-TRAINING DATA

Table F.1 presents a comparison of VoLTA on multimodal coarse-grained tasks with state-of-the-art methods pre-trained using magnitude more data. On VQA, VoLTA beat ViLBERT, UNITER-B, VILLA-B, UNIMO-B and ViLT-B, each pre-trained on $3-4$M datasets. Note that VoLTA is trained only on COCO and VG, whereas the other methods use a combination of COCO, VG, CC, and SBU datasets. Such strong performance proves the generalizability of VoLTA. On captioning, VoLTA beats Unified VLP, OSCAR, UFO-B, ViTCAP, VinVL-B, METER-CLIP-B and XGPT. However, for IRTR and NLVR, VoLTA can not yield better performance over these baselines. We assume that the large domain difference between pre-training and downstream datasets is the reason behind the limited performance on IRTR and NLVR.

# G    QUALITATIVE RESULTS

**Visual Question Answering and Visual Reasoning:** Visual question answering (VQA) is a widely recognized multi-modal task which infers an answer in response to a text-based question about an image. In Figure G.1 we demonstrated several example image-question pairs along with corresponding answers predicted by VoLTA on VQAv2 validation set. The primary aim of the visual reasoning task is to ascertain the veracity of a natural language statement against an associated image pair. Figure G.2 displays examples of responses (True/False) predicted by VoLTA on NLVR$^2$ validation set.

**Language-conditioned Object Detection:** Object detection forms an indispensable constituent of several multi-modal understanding systems. However, conventional object detection pipeline, being employed as a black-box tool, predicts all possible objects in the image. On the other hand, for better apprehension of combinations of these objects in free-form texts, language-conditioned object detection task is considered (Kamath et al., 2021; Dou et al., 2022a). We use pre-trained VoLTA for fine-tuning and evaluation on COCO and LVIS datasets for text-conditioned object detection task. As illustrated in the Figure G.3, VoLTA predicts bounding boxes relevant to the text prompts (captions)

Table F.1: **Multi-modal coarse-grained downstreams: visual question answering, visual reasoning, retrieval and captioning.** Methods pre-training with significantly larger dataset are colored gray. For captioning, 4 metrics are reported - B@4: BLEU@4, M: METEOR, C: CIDEr, S: SPICE. Best comparable results are in **bold**. VoLTA-B denotes Swin-B backbone.

| Method | #Pre-train Images | VQAv2 dev | std | NLVR² dev | test-P | F30k IRTR IR@1 | TR@1 |
|---|---|---|---|---|---|---|---|
| *Models pre-trained on COCO (123k) and/or VG (108k)* | | | | | | | |
| SCAN | 108k | – | – | – | – | 48.6 | 67.4 |
| SCG | 108k | – | – | – | – | 49.3 | 71.8 |
| PFAN | 108k | – | – | – | – | 50.4 | 70.0 |
| MaxEnt | 123k | 54.1 | 54.8 | – | – | – | – |
| VisualBERT | 123k | 70.8 | 71.0 | 67.4 | 67.0 | – | – |
| LXMERT | 231k | 72.4 | 72.5 | 74.9 | 74.5 | – | – |
| SOHO | 231k | 73.2 | 73.4 | 76.3 | 77.3 | 72.5 | **86.5** |
| *Models pre-trained on COCO, VG, SBU (1M) and/or CC (3M)* | | | | | | | |
| ViLBERT | 3M | 70.5 | 70.9 | – | – | 58.2 | – |
| UNITER-B | 4M | 72.7 | 72.9 | 77.2 | 77.9 | 72.5 | 85.9 |
| VILLA-B | 4M | 73.6 | 73.7 | 78.4 | 79.3 | 74.7 | 86.6 |
| UNIMO-B | 4M | 73.8 | 74.0 | – | – | – | – |
| ViLT-B | 4M | 71.3 | - | 75.7 | 76.1 | 64.4 | 83.5 |
| ALBEF-B | 4M | 74.5 | 74.7 | 80.2 | 80.5 | 82.8 | 94.3 |
| VLMo-B | 4M | 76.6 | 76.9 | 82.8 | 83.3 | 79.3 | 92.3 |
| METER-Swin-B | 4M | 76.4 | 76.4 | 82.2 | 83.5 | 79.0 | 92.4 |
| FIBER-B | 4M | 78.6 | 78.4 | 84.6 | 85.5 | 81.4 | 92.9 |
| VoLTA-B | 231k | **74.6** | **74.6** | **76.7** | **78.1** | **72.7** | 83.6 |

| Method | #Pre-train Images | COCO Captioning B@4 | M | C | S |
|---|---|---|---|---|---|
| *Models fine-tuned without CIDEr optimization* | | | | | |
| VL-T5 | 180k | 34.5 | 28.7 | 116.5 | 21.9 |
| VL-BART | 180k | 35.1 | 28.7 | 116.6 | 21.5 |
| Unified VLP | 3M | 36.5 | 28.4 | 117.7 | 21.3 |
| OSCAR | 4M | 36.5 | 30.3 | 123.7 | 23.1 |
| UFO-B | 4M | 36.0 | 28.9 | 122.8 | 22.2 |
| ViTCAP | 4M | 36.3 | 29.3 | 125.2 | 22.6 |
| METER-CLIP-B | 4M | 38.8 | 30.0 | 128.2 | 23.0 |
| VinVL-B | 4M | 38.2 | 30.3 | 129.3 | 23.6 |
| XGPT | 3.1M | 37.2 | 28.6 | 120.1 | 21.8 |
| FIBER-B | 4M | 39.1 | 30.4 | 128.4 | 23.1 |
| FIBER-GOLD-B | 4M | 40.3 | 30.7 | 133.6 | 23.6 |
| VoLTA-GOLD-B | 231k | 38.9 | 30.5 | 128.5 | 23.4 |
| *Models fine-tuned with CIDEr optimization* | | | | | |
| ViTCAP | 4M | 41.2 | 30.1 | 138.1 | 24.1 |
| VinVL-B | 4M | 40.9 | 30.9 | 140.4 | 25.1 |
| FIBER-B | 4M | 42.8 | 31.0 | 142.8 | 24.3 |
| FIBER-GOLD-B | 4M | 43.4 | 31.3 | 144.4 | 24.6 |
| VoLTA-GOLD-B | 231k | **40.2** | **30.9** | **137.5** | **23.7** |

and labels them with the corresponding spans from the text. For example, the top-middle image has 4 objects, however our model predicts boxes only for *person* and *cup* based on the text prompt.

**Referring Expression Comprehension (REC):** The objective of REC is to align the entire referring expression (text) with the corresponding box by disambiguating among the several occurrences of an object belonging to the same category and therefore, one box per expression is to be predicted. For example, bottom-left image in Figure G.4 depicts VoLTAś box prediction for the corresponding referring expression: *the slice of cake on the left*.

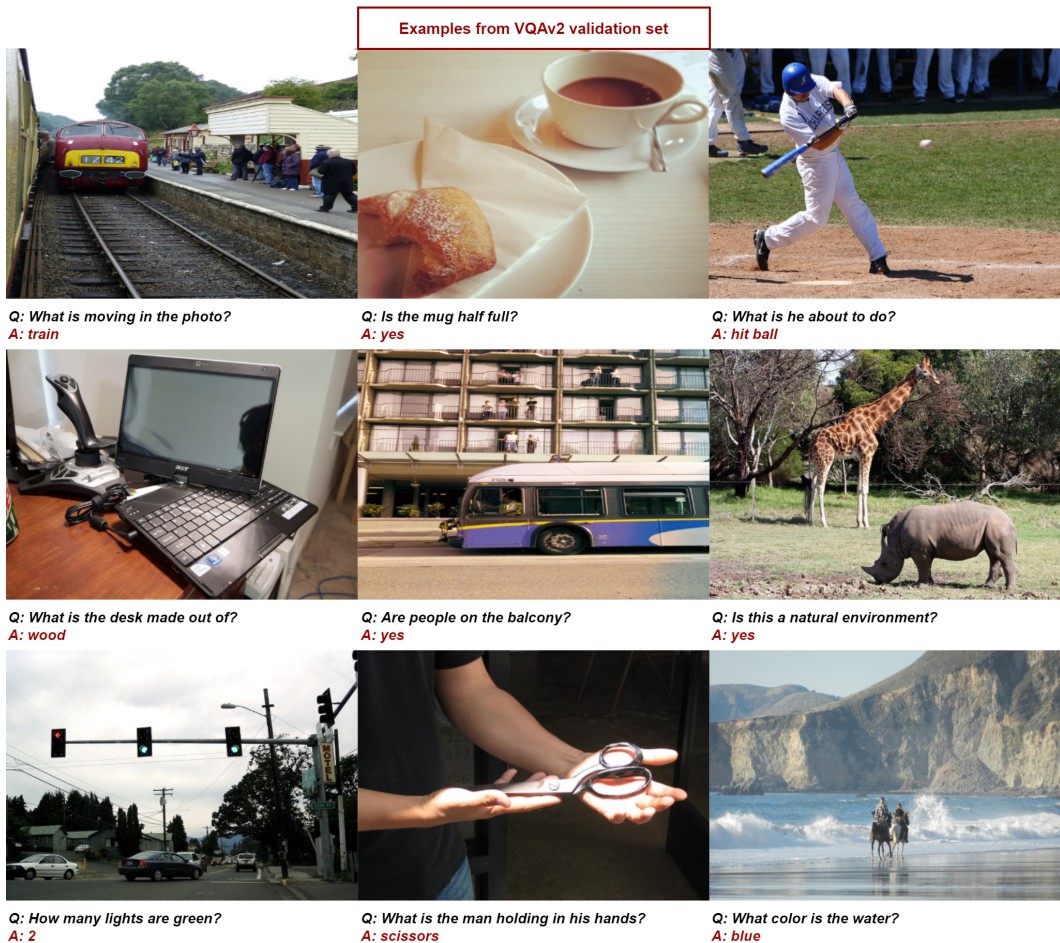

Figure G.1: **Examples on Visual Question Answering from VQAv2 validation dataset.** We display a variety of examples (e.g., number of items, color of objects, type of objects, events and actions) with respective answers predicted by VoLTA.

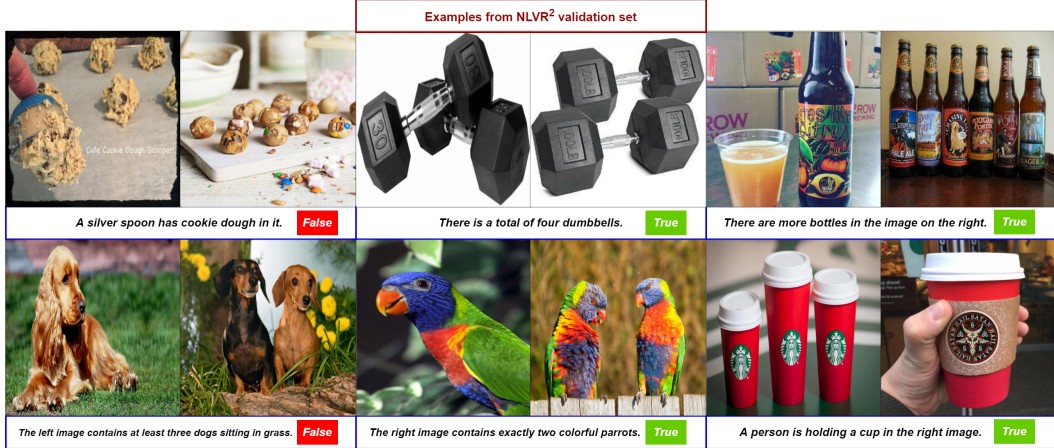

Figure G.2: **Examples on Visual Reasoning from NLVR$^2$ validation dataset.** For each statement (text prompt), 2 images are shown alongside each other and VoLTA predicts whether the given statement is True (green box) or False (red box).

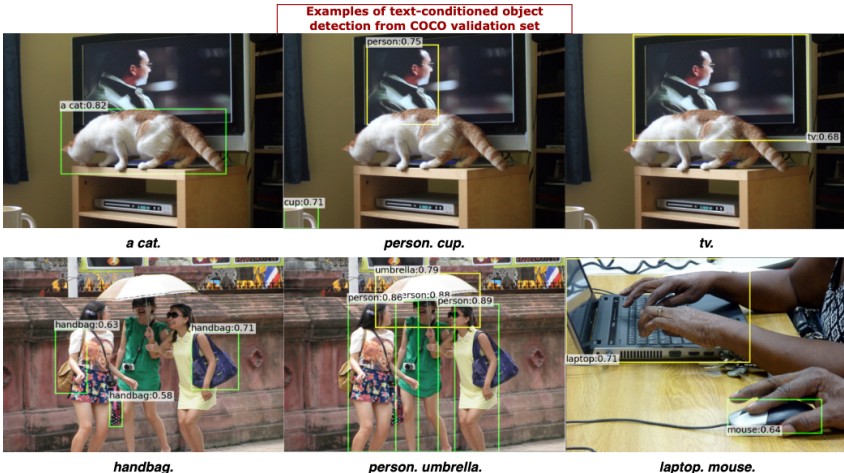

Figure G.3: **Examples of Object Detection from COCO validation dataset with various text prompts.** Our model predicts boxes relevant to the text (caption) and labels them with the corresponding spans.

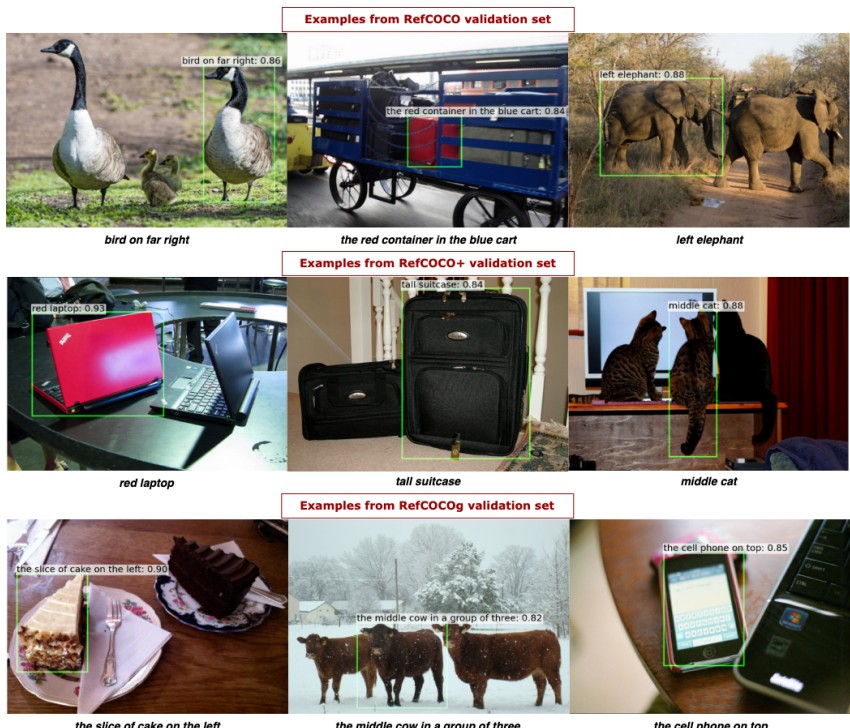

Figure G.4: **Examples of Referring Expression Comprehension from RefCOCO (top), RefCOCO+ (middle) and RefCOCOg (bottom) validation datasets.** The expressions in RefCOCOg typically have florid and longer constructions as compared to RefCOCO and RefCOCO+. The model has access to the entire text and uses it to disambiguate amongst different objects in the image.

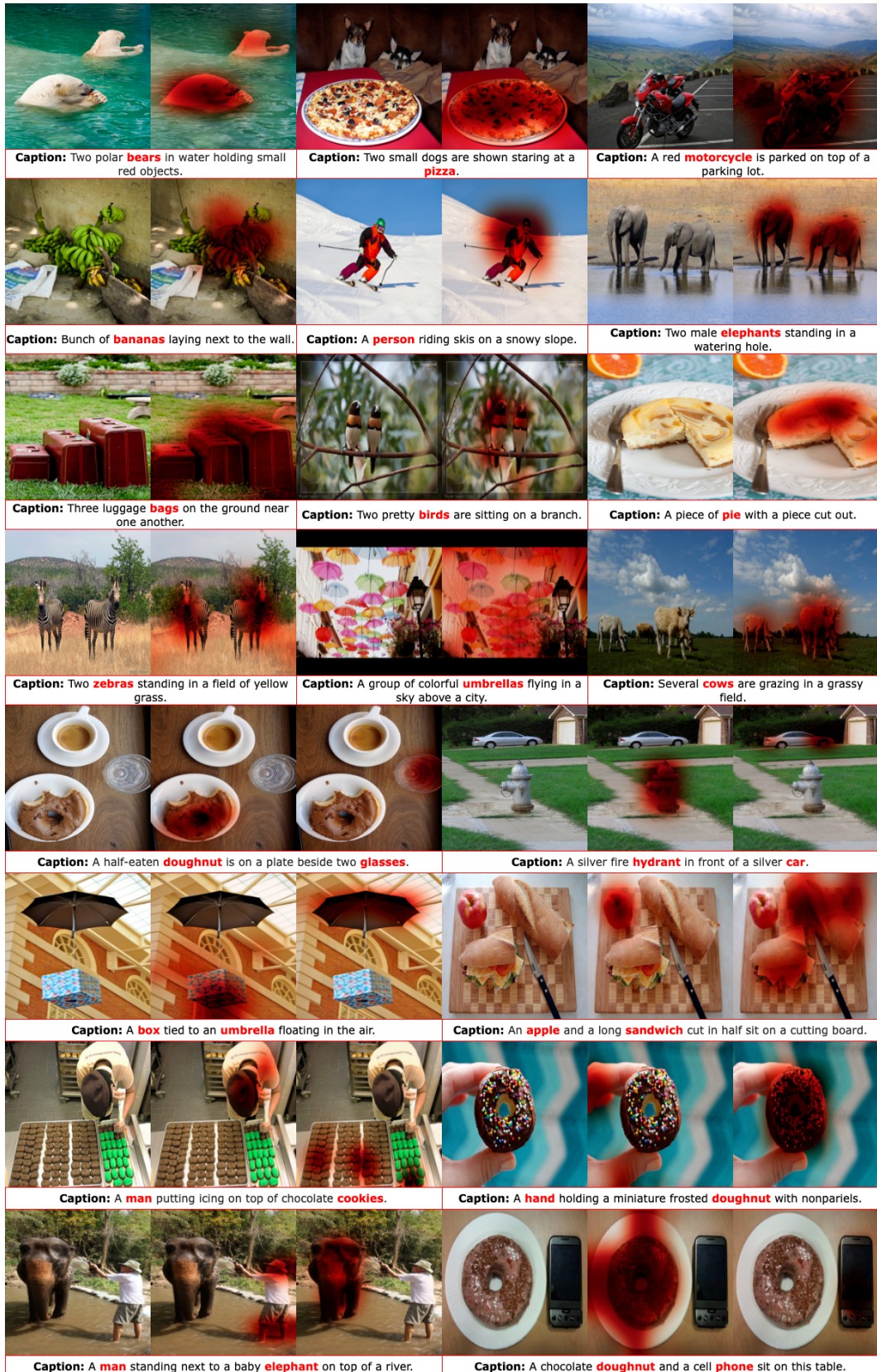

Figure G.5: **This figure shows how different words in captions attend relevant image regions, produced by the GOT module of VoLTA pre-trained on COCO.** Extension of Figure 1. All image-caption pairs are taken from the COCO2017 train split.

