# OpenReview forum: "VoLTA: Vision-Language Transformer with Weakly-Supervised Local-Feature Alignment"
_ICLR.cc/2023/Conference — Submitted to ICLR 2023_

### Official Review · Reviewer_PoWP · 2022-10-25

**Confidence:** 5
**Correctness:** 3
**Technical Novelty And Significance:** 1
**Empirical Novelty And Significance:** 2
**Recommendation:** 3

**Clarity, Quality, Novelty And Reproducibility:**

This paper is well-organized but misses some technical details (e.g., how GOT is applied). The technical novelty and contribution is not significant enough. Since the code is provided, the reproducibility is guaranteed.


**Strength And Weaknesses:**

Strength:
1. Using graph optimal transport (GOT) in multimodal problems is interesting, although the advantage of GOT vs other alignment methods is not investigated (see more details in weakness).
2. The paper is well-organized and easy to follow.
3. Thanks for providing the code which can help readers to reproduce the results.

Weakness:
1. This paper extends BT to VLP problems given the fact that BT does not require large batch size. What if contrastive-based methods use the same batch size as BT, will it degrade the performance significantly? If not, then we can simply use small batch sizes.
2. How was GOT applied to image features and text features? This is not clearly discussed in the Method section.
3. On page 5: WD can only match nodes in the graph, and will treat all “men” entities as identical and will ignore neighbouring relations like “in blue shirt” and “holding the scissors”. Is there any evidence to support this statement?
4. Intra-modal objective and inter-modal objective share the similar idea of TCL [1], except that TCL is built upon contrastive losses. Furthermore, TCL is a very important work in VLP.
5. In Equation 2 and 3, what is $k’$ in $C_{i,i}^{k’}$?
6. What does “weakly-supervised” represent in this paper?
7. On page 7: 	“For MLM, we randomly mask 15% text tokens”. Have you replaced the masked tokes with other special tokens?
8. What’s the advantage of using GOT compared with existing local alignment methods such as OT?
9. The performance improvement is marginal by using the same backbones.

[1] Vision-Language Pre-Training with Triple Contrastive Learning, CVPR 2022


**Summary Of The Paper:**

This paper proposes a new vision-language pre-training (VLP) method for local feature alignment. The key idea is to use a graph optimal transport mechanism to align image patches and text tokens. This paper also extends BT to VLP by introducing an intra-modal objective and an inter-modal objective. Finally, a gated cross-attention is designed to eliminate the fusion encoder that is widely used in previous work.


**Summary Of The Review:**

Although this paper extends existing methods to the VLP field, the motivation is not clear. Furthermore, the advantages of the proposed modules are not investigated or empirically proved. The technical novelty/contribution and performance improvement is quite marginal.

---

> ### Author Response · Authors · 2022-11-14
> **Response to Reviewer PoWP (Part 1/2)**
>
> **Q1: This paper extends BT to VLP problems, given that BT does not require a large batch size. What if contrastive-based methods use the same batch size as BT? Will it degrade the performance significantly?**
>
> We thank the reviewer for asking this fundamental question. It is important to note that contrastive objectives suffer substantially with small batch sizes. This phenomenon is widely discussed in the literature [13,17,18,19]. Such happens because contrastive methods heavily rely on in-batch negative samples. With an increase in batch size, a diverse collection of negative samples can be covered, improving the system’s robustness by learning meaningful representations to distinguish positive and hard-negative examples. The impact of batch size on contrastive algorithms’ performance has been clearly illustrated in Figure 3a by Yeh et al. [17] and Figure 9 by Chen et al. [18].
>
> Moreover, Pham et al. [19] provided theoretical insights on the impact of batch sizes on contrastive vision-language pre-training (VLP). In particular, the generalization gap of a model is inversely related to the batch size. Pham et al. mathematically showed that this generalization gap decreases at the rate of $\mathcal{O}(\frac{1}{\sqrt{m}} + \frac{1}{\sqrt{B}})$ where $m$ is the number of training samples, and $B$ is the batch size (Theorems 1 and 2 of [19]). However, with a large number of training samples ($m$), if $B$ is small, the generalization gap can still be significant. Therefore, to reduce this performance gap, $B$ needs to be high. Pham et al. experimentally validated their theoretical claims by conducting ablation studies with different batch sizes and demonstrated that contrastive learning benefits from higher batch sizes (Table 4 of Pham et al. [19]). In particular, when batch size was decreased from 32768 to 4096, top-1 zero-shot accuracy on ImageNet decreased by ~4\% for their small model (BASIC-S) and ~5\% for their medium model (BASIC-M).
>
> On the contrary, non-contrastive objectives, such as Barlow Twins (BT) [13], are robust to small batch sizes. For example, Figure 2 in [13] shows that the top-1 accuracy difference on ImageNet (linear evaluation) between batch sizes of 256 and 2048 is ~3.5\% for SimCLR [18], whereas it is ~0.7\% for BT [13]. These observations motivated us to use non-contrastive loss (such as BT) for a more efficient and robust multi-modal training paradigm.
>
> **Q2: How was GOT applied to image features and text features?**
>
> We are sorry for the confusion. We obtain the patch- and token-level features from the last layers of corresponding visual and textual transformer encoders and use these encoded local-feature vectors to construct modality-specific dynamic graphs - $\mathcal{G}_x(\mathbb{V}_x, \mathcal{E}_x)$ for image patches and $\mathcal{G}_y(\mathbb{V}_y, \mathcal{E}_y)$ for text tokens. Each node in these graphs $i \in \{\mathbb{V}_x, \mathbb{V}_y\}$ is represented by corresponding feature vectors, and intermediate edges $e \in \{\mathbb{E}_x, \mathbb{E}_y\}$ by thresholded cosine similarity. Once $\mathcal{G}_x$ and $\mathcal{G}_y$ are computed, we follow the GOT algorithm [9] to compute WD and GWD. We explain this alignment strategy in section 3.1.2 and our pseudo code (Appendix A).
>
> **Q3: On page 5: WD can only match nodes in the graph, will treat all “men” entities as identical, and will ignore neighboring relations like “in blue shirt” and “holding the scissors.” Is there any evidence to support this statement?**
>
> We are sorry for the confusion. We provide detailed reasons why WD can not distinguish between similar entities represented by other nodes in the same graph in “Response to All Reviewers” Q4 and in section 3.1.2 of our manuscript.
>
> To summarize, the definition of WD in Equation 4 in the manuscript shows that WD is optimized for matching nodes in different graphs, failing to capture similarities between edges. Let us consider a simple example from the original GOT [9] paper. Given the sentence “there is a red book on the blue desk” paired with an image containing several desks and books in different colors, WD can not correctly distinguish which colored book and desk the sentence refers to. Hence, WD loses neighboring node information during alignment and fails to understand the relationships among similar objects in the image. The same argument applies to the example we provided in Figure 3 of our manuscript.
>
> [9] Liqun Chen et al.; Graph optimal transport for cross-domain alignment. ICML, 2020.
>
> [13] Jure Zbontar et al.;. Barlow twins: Self-supervised learning via redundancy reduction. ICML, 2021.
>
> [17] Chun-Hsiao Yeh et al.; Decoupled contrastive learning. arXiv, 2021.
>
> [18] Ting Chen et al.; A simple framework for contrastive learning of visual representations. ICML, 2020.
>
> [19] Hieu Pham et al.; Combined scaling for open-vocabulary image classification. arXiv, 2021.

---

> > ### Author Response · Authors · 2022-11-14
> > **Response to Reviewer PoWP (Part 2/2)**
> >
> > **Q4: Intra- and inter-modal objectives share a similar idea of TCL [5], except that TCL is built upon contrastive losses.**
> >
> > We thank the reviewer for mentioning this important reference. Please find the detailed motivation of this work in “Response to All Reviewers” Q1, which explains how this work differs from the wide range of existing vision-language pre-training (VLP) literature. We agree that intra- and inter-modal contrastive objects have previously been used in the literature [5,11,12]. However, we are the first to extend this concept to non-contrastive domains.
> >
> > We indicate that TCL [5] uses cross-modal alignment on global image-text features using [CLS] tokens from the two modalities. As opposed to this, VoLTA proposes region-level image-text alignment using local features (image patches and text tokens) obtained from respective encoders. We leverage graph optimal transport to accomplish this local alignment task using only global image-caption pairs without any expensive image-text box annotation. Moreover, TCL has not reported its performance on fine-grained region-level tasks, such as object detection and REC. We also highlight that VoLTA is one of the few works to address such region-level alignment using global image-text pairs.
> >
> >
> > **Q5: In Equation 2 and 3, what is $k′$ in $C^{k'}_{ii}$?**
> >
> > We thank the reviewer for indicating this typo. The $k’$ should be changed to $k$. We have made the required changes in Equation 2 and 3.
> >
> >
> > **Q6: What does “weakly-supervised” represent in this paper?**
> >
> > We are sorry for the confusion. Please find our detailed answer to this question in “Response to All Reviewers” Q1.
> >
> >
> > **Q7: On page 7: “For MLM, we randomly mask 15% text tokens”. Have you replaced the masked tokes with other special tokens?**
> >
> > We thank the reviewer for asking about this important masking detail. We are sorry that the last version of our manuscript did not mention this detail. Following the literature [6, 20], we randomly mask out the input words with a probability of 15\% and replace the 80\% masked words with a special token [MASK], 10\% with random words, and the rest 10\% the original words. We have included these details in **Appendix Section D.2.**
> >
> > **Q8: What’s the advantage of using GOT compared with existing local alignment methods such as OT?**
> >
> > We thank the reviewer for asking this important question. Please find our detailed answer to this question in “Response to All Reviewers” Q4 and “Response to Reviewer mig1” Q3.
> >
> >
> > **Q9: The performance improvement is marginal by using the same backbones.**
> >
> > We thank the reviewer for asking this important question. We highlight that VoLTA achieves state-of-the-art results across most metrics on all 6 unimodal downstream tasks. Since the primary strength of our system is on region-level tasks, VoLTA achieves superior performance on object detection and instance segmentation compared to all baselines, even when trained using ten times fewer pre-training samples. For coarse-grained unimodal tasks, VoLTA also achieves state-of-the-art performance on image classification on ImageNet, VOC07, and COCO. Please note that we can directly infer the difficulty of all these tasks from the very close scores achieved by all strong baselines. We discuss the desired efficacy of our system on multi-modal tasks in “Response to All Reviewers” Q3 and in Section 4.2 and Section 4.3.
> >
> > [5] Jinyu Yang et al.; Vision-Language Pre-Training with Triple Contrastive Learning. CVPR, 2022.
> >
> > [6] Yen-Chun Chen et al.; UNITER: UNiversal Image-TExt Representation Learning. ECCV, 2020.
> >
> > [11] Yangguang Li et al.; Supervision exists everywhere: A data efficient contrastive language-image pre-training paradigm. ICLR, 2022.
> >
> > [12] Xin Yuan et al.; Multi-modal contrastive training for visual representation learning. CVPR, 2021.
> >
> > [20] Jacob Devlin et al.; Bert: Pre-training of deep bidirectional transformers for language understanding. NAACL, 2019.

---

> ### Author Response · Authors · 2022-11-18
> **Has our response addressed your concerns?**
>
> Dear Reviewer PoWP:
>
> We would be grateful if you kindly confirm whether our response has addressed your concerns and let us know if any issues remain. In the following, we recap the key points of our response:
>
> 1. With adequate references from the literature, we thoroughly explained the reasons why the performance of contrastive algorithms degrades with smaller batch sizes. On the other hand, non-contrastive algorithms (such as BT) are more robust on batch size.
> 2. We clarified the efficacy of utilizing WD and GWD mutually beneficially in a joint transport plan for accurate patch-word matching. The advantage of using such a transport algorithm (GOT) is also illustrated with additional ablation experiments.
> 3. We explained how the patch- and token-level features from the last layers of corresponding visual and textual transformers are used in the GOT algorithm for accurate image-text matching.
> 4. We recapitulated the detailed motivation behind VoLTA and explained how this work differs from the wide range of existing vision-language pre-training (VLP) literature. We also explicitly pointed out how VoLTA is different from TCL.
> 5. We clearly explained the meaning of “weakly-supervised” in this paper.
> 6. We fixed typos in Equations 2 and 3.
> 7. We mention the masking strategy for MLM. Following existing literature, we randomly mask out the input words with a probability of 15% and replace the 80% masked words with a special token [MASK], 10% with random words, and the rest 10% with the original words.
> 8. We explained the significance of the performance of our proposed system with a wide range of downstream and ablation experiments.
>
> We appreciate your constructive feedback, which allowed us to improve our paper. We would be grateful if you kindly consider increasing the score. We look forward to your feedback!

---

### Official Review · Reviewer_mig1 · 2022-10-26

**Confidence:** 3
**Correctness:** 2
**Technical Novelty And Significance:** 2
**Empirical Novelty And Significance:** 2
**Recommendation:** 5

**Clarity, Quality, Novelty And Reproducibility:**

The idea of this paper is interesting, but the figures, evaluation of the proposed methods are not clear.

**Strength And Weaknesses:**

Strength:
1.	Many experiments on downstream tasks are conducted.
2.	Paper writing and structure are easy to understand.
3.	The idea two views of image and text in GOT part is interesting.

Weakness:
1.	The figures are not clear. For example, in figure 2, it’s confused for the relation of 3 sub-figures. Some modules are not labeled in figure, such as CMAF, L_BT, VoLTA.
2.	The experiments results are not significant.
3.	Three steps for training are shown in VoLTA, a) switching off CMAF, b) switching on CMAF, c) keep CMAF and random sampling for training. The ablation study on these parts should be conducted.
4.	The key point of this paper is GOT, but no ablation on this part. The authors are encouraged to verify which parts works.


**Summary Of The Paper:**

In this paper, the authors propose VoLTA, a unified Vision Language Pre-training paradigm for image-level and region-level applications, in which only image-caption pairs are used. They use graph optimal transport for weakly-supervised feature-level and region-level alignment. They conduct a wide range of vision- and vision-language downstream experiments to demonstrate the effectiveness of VoLTA.

**Summary Of The Review:**

This paper proposes a new VLP framework using GOT, but the evaluation is in-sufficient and the effectiveness of the proposed module is not clear.

---

> ### Author Response · Authors · 2022-11-14
> **Response to Reviewer mig1**
>
> **Q1: In Figure 2, the connection across three submodules needs to be clarified.**
>
> We cordially thank the reviewer for raising this critical issue. We agree that the last version of Figure 2 was suboptimal, and the three submodules seemed isolated. **We have modified Figure 2** substantially to indicate the cross-attention mechanism in our system. During the computation of the barlow twins (BT) and graph optimal transport (GOT) loss, we switch off the gated cross-attention; and during masked language modeling and image-text matching loss, we switch on the gated cross-attention. Thus, the gated cross-attention mechanism in VoLTA allows us to flexibly switch between dual and fusion encoder in every forward pass during pre-training. We also clearly indicate how the image-text pair is modified for the computation of different losses. Lastly, we add the missing labels into this figure.
>
> **Q2: The effectiveness of the proposed system is insignificant.**
>
> We thank the reviewer for asking the significance of the performance of VoLTA. Please find our detailed answer to this question in “Response to All Reviewers” Q3.
>
> To summarize, the primary efficacy of VoLTA lies in the fact that VoLTA does not use any image-text-box data yet achieve superior performance on fine-grained downstream tasks, such as language-conditioned object detection and referring expression comprehension (REC). We can see in Table 4 that VoLTA beats all base-sized and larger models trained without image-text-box data across all validation and test sets of RefCOCO, RefCOCO+, and RefCOCOg. Moreover, VoLTA performs comparably with MDETR and UniTAB, even without being trained on grounding data. Finally, table 5 shows that VoLTA outperforms several strong baselines trained with grounding data on LVIS. These results prove that our system can achieve impressive localization capability by only utilizing a global image-caption corpus.
>
>
> **Q3: The ablation study on different pre-training objectives should be conducted. The key point of this paper is GOT, but no ablation on this part. The authors are encouraged to verify which parts work.**
>
> We thank the reviewer for raising this important point. We agree that some crucial ablation experiments were missing in our paper. Please find our detailed answer to this question in “Response to All Reviewers” Q4.
>
> Table E.2. analyzes the effectiveness of different pre-training objectives through an ablation on fine-grained downstream tasks. This table also ablates the two types of OT distances used in GOT - WD for node matching and GWD for edge matching. As shown in Table E.2, the GWD loss helps improve reference expression comprehension performance across RefCOCO, RefCOCO+, and RefCOCOg datasets. Specifically on RefCOCOg, adding the GWD loss yields a significant 4.0\% boost in the challenging test set. This dataset contains intricate images with multiple similar objects with different shapes and colors; hence GWD is crucial in distinguishing between them. Overall, this set of ablation experiments demonstrates that all of the objectives are necessary for our model to obtain good performance on different fine-grained multi-modal tasks.

---

> ### Author Response · Authors · 2022-11-18
> **Has our response addressed your concerns?**
>
> Dear Reviewer mig1:
>
> We would be grateful if you kindly confirm whether our response has addressed your concerns and let us know if any issues remain. In the following, we recap the key points of our response:
>
> 1. We have substantially modified Figure 2 to indicate the connections between three submodules. We also distinctly demonstrate how an image-text pair is adjusted for different loss computations.
> 2. We clarified the motivation of VoLTA and minutely explained the significance of its performance.
> 3. We described the efficacy of GOT for the alignment of intricate image-caption pairs. We conducted detailed ablation experiments to showcase the usefulness of GOT over traditional OT algorithms in VLP. Moreover, we also demonstrated that all objectives in VoLTA are necessary to obtain good performance on different fine-grained multi-modal tasks.
>
> We appreciate your constructive feedback, which allowed us to improve our paper. We would be grateful if you kindly consider increasing the score. We look forward to your feedback!

---

### Official Review · Reviewer_RqXJ · 2022-11-04

**Confidence:** 4
**Clarity, Quality, Novelty And Reproducibility:** 1. The paper has good clarity on a hi…
**Correctness:** 4
**Technical Novelty And Significance:** 4
**Empirical Novelty And Significance:** 3
**Recommendation:** 6

**Strength And Weaknesses:**

Strength -

(+) The concepts and higher level overview of the problem is somewhat well defined.

(+) The idea of using Barlow-Twins for both the Image and Text modality and thereby performing inter-modal and intra-modal redundancy reduction is well appreciated and interesting.

(+) The idea of using Graph Optimal Transport instead of Optimal Transport for patch-word alignment seems intuitive and novel for this vision-language domain.

(+) Extensive experiments have been performed on multiple standard benchmarks often surpassing supervised approaches.

Weakness -

(-) The reviewer is very curious about the textual augmentations. The amount of augmentations done on the textual caption looks like a strong data augmentation and in such cases Barlow-Twins may not perform well. Can the authors shed some light on this issue ? This might affect the overall model performance while doing the intra-modal alignment.

(-) The cross-modal attention fusion block part appeared a bit unclear to this reviewer. How is this CMAF module trained ? When alpha=0, only BT and GOT losses are computed , however when non zero then MLM and ITM is computed, does that mean , this is trained in a 2-stage fashion ? i.e first the BT and GOT loss is computed to get a good initilaization point and then CMAF is activated to do the fusion ? This needs some clarification.

(-) This reviewer is also curious to know how is the inference on dense downstream task like object detection and instance segmentation performed ? After VolTa pre-training, is it fine-tuned on the target dataset or the decoders are attached to the CMAF only for finetuning ? This needs some clarification.

(-) What is the purpose of such pre-training in general ? Since the method performs very well on uni-modal downstream tasks but lags behind in multi-modal downstream tasks, it is hard to understand the motivation for such pre-training. The reviewer agrees that annotations are costly, and such weak supervision provides an alternate solution, but only surpassing uni-modal downstream tasks using multi-modal pre-training looks to this reviewer a slightly weaker claim.

(-) Why is the gating based attention is better than traditional cross-attention transformer. Such ablations should be done to justify the design choice which is missing in current form.

(-) The paper has minor grammatical errors which should be fixed for better readibility

**Summary Of The Paper:**

The paper proposes a new Vision-language pre-training (VLP) paradigm that only utilizes image-caption data but achieves fine-grained region-level image understanding, eliminating the use of expensive box annotations. Existing end-to-end  VLP methods use high-resolution image-text-box data to perform well on finegrained region-level tasks, such as object detection, segmentation, and referring expression comprehension. However, it comes at high annotation cost. As a solution to this problem, the authors proposed VoLTA which uses adopts graph optimal transport-based weakly-supervised alignment on local image patches and text tokens to germinate an explicit, self-normalized, and interpretable low-level matching criterion. As a result, VoLTA pushes multi-modal fusion deep into the uni-modal backbones during pre-training and removes fusion-specific transformer layers, further reducing memory requirements.  Extensive experiments on a wide range of vision- and vision-language downstream tasks demonstrate the effectiveness of VoLTA on fine-grained applications without compromising the coarse-grained downstream performance.

**Summary Of The Review:**

Although i think the whole idea is interesting and worthy of discussion in the research community, i am a bit concerned at the use-case of this pre-training as compared to let say off-the-shelf GLIPv2. However, I agree with the authors that such pre-training is indeed necessary in cases where annotations are not available and can provide some useful insights in the near-future. Regarding the draft, the reviewer is slightly confused in how the fusion module is trained and what is the imapct of the textual augmentations in Barlow Twins. Some minor issues like grammatical errors and missing ablations also need some attention. Therefore, at this stage, I will vote for marginal accept and re-evaluate based on authors and other reviewer response.

---

> ### Author Response · Authors · 2022-11-14
> **Response to Reviewer RqXJ (Part 1/2)**
>
> **Q1: How is the training of the cross-modal attention fusion (CMAF) module?**
>
> We thank the reviewer for requesting clarification on this essential training detail. Please find our detailed answer to this question in “Response to All Reviewers” Q2.
>
> In summary, each forward pass of our pre-training is composed of three steps - (i) We switch off the gate; hence CMAF is turned off, and VoLTA works as a dual encoder to compute BT and GOT losses. (ii) We switch on the gate, VoLTA works as a fusion encoder to compute MLM, and (iii) ITM loss. After the three steps of the forward pass are completed, all four losses are added and used for back-propagation. The gated-attention mechanism in VoLTA allows us to flexibly switch between dual and fusion encode in every forward pass during pre-training.
>
> We explain our detailed training strategy in section 3.1.3 and our pseudo code (Appendix A). **We have also modified Figure 2** to clarify the connection between different modules in our system.
>
> **Q2: What is the impact of the textual augmentations in barlow twins? Does such augmentation affect inter-modal alignment?**
>
> We thank the reviewer for raising this important question. We use the EDA [10] method as our text augmentation strategy, which contains four text augmentations: synonym replacement, random insertion, random swap, and random deletion. Such EDA-based text augmentation has been previously used in contrastive VLP literature [11, 12]. We have kept the probability of each four augmentations sufficiently low so that the original semantics of the caption is preserved. Hence, such augmentation does not affect inter-modal alignment. We present all augmentation details in Table D.1.
>
> We also verify the effectiveness of the multi-modal barlow twins (BT) objective by ablating the intra- and inter-modal terms. See the results below:
>
> Table E.3: **Ablation study on Intra- and Inter-modal Barlow Twins objective for multi-label image classification on VOC07 and COCO.** We report classification mAP on VOC$07$, and per-class (PC) and overall (O) F$1$ scores on COCO. Each model is pre-trained on $123$k train-val samples from COCO$2017$.
>
> | $\mathcal{L}_{\mathrm{BT}}^{II'}$ |  $\mathcal{L}_{\mathrm{BT}}^{TT'}$ |  $\mathcal{L}_{\mathrm{BT}}^{IT'}$ |  $\mathcal{L}_{\mathrm{BT}}^{I'T}$ |  VOC07 | COCO (PC/O) |
> | :----: | :----: | :-----: | :-----: | :----: | :----: |
> | &check; | − | − | − | 90.8 | 72.2/71.7 |
> | &check; | &check; | − | − | 91.8 | 74.0/73.7 |
> | &check; | &check; | &check; | &check; | **94.0** | **74.5/76.4** |
>
>
> The first row of this table is identical to the original image-only BT objective [13]. Next, we introduce the text branch and add the BT objective between the two views of the caption. Afterward, we add the inter-modal BT objectives. As shown in the table, each loss term improved the image classification performance, demonstrating the importance of intra- and inter-modal objectives.
>
> We added this result in the **Appendix Table E.3.**
>
> **Q3: How is the inference on dense downstream tasks like object detection and instance segmentation performed? After VoLTA pre-training, are the backbones fine-tuned on the target dataset or are only the task-specific heads fine-tuned?**
>
> We thank the reviewer for asking about this necessary fine-tuning detail. We describe our detailed fine-tuning strategy in Section 3.2 of the manuscript. During the vision-only tasks, such as image classification, object detection, and instance segmentation, we switch off the CMAF module and use the image encoder. Following the standard experimental protocol used in the literature [13, 14, 15, 16], we only fine-tune the classification heads for image classification and the entire backbone for object detection and instance segmentation. **We have clarified these essential details in the caption of Table 1 and Table 2.**
>
> [10] Jason Wei et al.; EDA: easy data augmentation techniques for boosting performance on text classification tasks. EMNLP, 2019.
>
> [11] Yangguang Li et al.; Supervision exists everywhere: A data efficient contrastive language-image pre-training paradigm. ICLR, 2022.
>
> [12] Xin Yuan et al.; Multi-modal contrastive training for visual representation learning. CVPR, 2021.
>
> [13] Jure Zbontar et al.; Barlow twins: Self-supervised learning via redundancy reduction. ICML, 2021.
>
> [14] Kaiming He et al.; Momentum contrast for unsupervised visual representation learning. CVPR, 2020.
>
> [15] Mathilde Caron et al.; Unsupervised learning of visual features by contrasting cluster assignments. NeurIPS, 2020.
>
> [16] Jean-Bastien Grill et al.; Bootstrap your own latent: A new approach to self-supervised learning. NeurIPS, 2020.

---

> > ### Author Response · Authors · 2022-11-14
> > **Response to Reviewer RqXJ (Part 2/2)**
> >
> > **Q4: What is the purpose of such pre-training in general?**
> >
> > We thank the reviewer for requesting elucidation on the motivation of our work. Please find our detailed answer to this question in “Reply to All Reviewers” Q1. We have also written the detailed significance of VoLTA in “Reply to All Reviewers” Q3.
> >
> > **Q5: Why is the gating-based attention better than the traditional cross-attention transformer?**
> >
> > Please find our detailed answer to this question in “Reply to All Reviewers” Q2.
> >
> > The advantage of using a gated-attention mechanism over traditional cross-attention transformer layers is twofold: (i) flexibility of switching between dual and fusion encoders. In addition, (ii) the gating mechanism is more lightweight and memory-efficient than fusion-specific layers (GLIP and METER use 110M fusion parameters; VoLTA uses 26M fusion parameters for base-sized models).

---

> ### Author Response · Authors · 2022-11-18
> **Has our response addressed your concerns?**
>
> Dear Reviewer RqXJ:
>
> We would be grateful if you kindly confirm whether our response has addressed your concerns and let us know if any issues remain. In the following, we recap the key points of our response:
>
> 1. We clarified the training strategy of cross-modal attention fusion (CMAF) and explained the benefits of using gated-attention over traditional cross-attention transformer layers.
> 2. We described the impact of the textual augmentations in Barlow Twins and clarified why our textual augmentations did not affect inter-modal alignment.
> 3. We clarified the detailed inference strategies on dense downstream tasks such as object detection and instance segmentation.
> 4. We reiterated the motivation behind VoLTA and explained how this work differs from the wide range of existing vision-language pre-training (VLP) literature. We also explicitly pointed out how VoLTA differs from existing off-the-shelf methods, like GLIPv2.
>
> We appreciate your constructive feedback, which allowed us to improve our paper. We would be grateful if you kindly consider increasing the score. We look forward to your feedback!

---

### Author Response · Authors · 2022-11-14
**General Response to All Reviewers (Part 1/4)**

We thank the reviewers for their constructive feedback, which allowed us to improve our paper. We appreciate that all reviewers find our approach insightful, novel, and easy to follow. However, we also admit the need for clarity in explaining the necessity of gated attention and more ablation experiments to reinforce the advantages of using graph-optimal transport (GOT) for achieving fine-grained image-text alignment using global image-caption annotations. Moreover, we also scale up our pre-training on larger datasets, significantly improving the performance across all multimodal downstream tasks. **All the modifications in the manuscript are colored blue.**

We believe that our in-depth explanation, additional pre-training and ablation experiments will help to clarify the raised concerns, and we kindly ask the reviewers to consider increasing their scores. We have updated the paper accordingly.

Here, we address a few common questions and demonstrate additional experiments. We also specifically respond to each reviewer for detailed questions.

**Q1. Although this paper extends existing methods to the VLP field, the motivation is not clear.**

We first reiterate the motivation behind VoLTA and explain how this work differs from the wide range of existing vision-language pre-training (VLP) literature. This illustration also addresses the concern of reviewer RqXJ by pointing out that VoLTA is different from existing off-the-shelf methods, like GLIPv2 [1].

Traditionally, most existing end-to-end VLP methods [1,2,3] with region-level understanding exploit high-resolution image-text-box pre-training data to perform well on fine-grained region-level tasks, such as object detection and referring expression comprehension (REC). However, accurate bounding box annotations on high-resolution input images are expensive to collect, annotate and use for pre-training at scale. Recently, FIBER [4] addressed this vital problem and proposed a two-stage pre-training paradigm to lessen the use of expensive bounding box annotations. **In this work, we move a step forward to eliminate box annotations completely but achieve comparable performance on fine-grained region-level tasks.**

Our motivation also addresses reviewer PoWP to explain the meaning of “weakly-supervised” in this paper. VoLTA uses global image-caption annotations, but achieves region-level fine-grained understanding by weakly-supervised alignment of image patches and text tokens. Moreover, we also address reviewer PoWP by indicating that TCL [5] uses cross-modal alignment on global image-text features using [CLS] tokens from the two modalities. As opposed to this, VoLTA proposes region-level image-text alignment using local features (image patches and text tokens) obtained from respective encoders. We leverage graph optimal transport to accomplish the local alignment task using only global image-caption pairs without any expensive image-text box annotation. Moreover, TCL has not reported its performance on fine-grained region-level tasks, such as object detection and REC. We also highlight that VoLTA is one of the few works to address such region-level alignment using global image-text pairs.

[1] Haotian Zhang et al.; Glipv2: Unifying localization and vision-language understanding. NeurIPS, 2022.

[2] Liunian Harold Li et al.; Grounded language-image pre-training. In CVPR, 2022.

[3] Aishwarya Kamath et al.; Mdetr-modulated detection for end-to-end multi-modal understanding. CVPR, 2021.

[4] Zi-Yi Dou et al.; Coarse-to-fine vision-language pre-training with fusion in the backbone. NeurIPS, 2022.

[5] Jinyu Yang et al.; Vision-Language Pre-Training with Triple Contrastive Learning. CVPR, 2022.

---

> ### Author Response · Authors · 2022-11-14
> **General Response to All Reviewers (Part 2/4)**
>
> **Q2: How is the cross-modal attention fusion (CMAF) module trained? What is the benefit of such a gated-attention mechanism over traditional cross-attention transformer layers?**
>
> We thank the reviewers (Reviewer RqXJ and Reviewer mig1) for requesting clarification on this important architectural detail, which is the key to significantly reducing the number of trainable fusion parameters.
>
> Broadly, vision-language systems are of two types - dual encoder and fusion encoder. Dual encoder systems are usually trained on image-text contrastive (ITC) loss [CLIP loss]. In contrast, fusion encoders utilize a wide range of masking/matching losses - masked language modeling (MLM), masked image modeling (MIM), and image-text matching (ITM).
>
> In contrast to most existing literature, our proposed system, VoLTA, is flexible enough to switch between dual and fusion encoder architecture. The dual encoder is used to compute the BT and GOT losses, whereas the fusion encoder is used for MLM and ITM losses. The gated-attention mechanism achieves such flexibility. Each forward pass of our pre-training is composed of three steps - (i) We switch off the gate; hence CMAF is turned off, and VoLTA works as a dual encoder to compute BT and GOT losses. (ii) We switch on the gate, VoLTA works as a fusion encoder to compute MLM, and (iii) ITM loss. After the three steps of the forward pass are completed, all four losses are added and used for back-propagation. We explain this gated-attention training strategy in section 3.1.3 and our pseudo code (Appendix A).
>
> The advantage of using a gated-attention mechanism over traditional cross-attention transformer layers is twofold: (i) flexibility of switching between dual and fusion encoders. In addition, (ii) the gating mechanism is more lightweight and memory-efficient than fusion-specific layers (GLIP and METER use 110M fusion parameters; VoLTA uses 26M fusion parameters for base-sized models).

---

> > ### Author Response · Authors · 2022-11-14
> > **General Response to All Reviewers (Part 3/4)**
> >
> > **Q3: How effective is the proposed system?**
> >
> > We thank the reviewers (Reviewer RqXJ and Reviewer mig1) for asking this pivotal question. The primary efficacy of VoLTA lies in the fact that VoLTA does not use any image-text-box data yet achieve superior performance on fine-grained downstream tasks, such as language-conditioned object detection and referring expression comprehension (REC). We also show the flexibility of VoLTA by benchmarking on a wide range of unimodal and coarse-grained multimodal downstream tasks.
> >
> > In our existing manuscript, we pre-trained VoLTA on the image-caption pairs of the COCO2017 split, which consists of 123k images. During the past 44 days, as we received ample time and resources, following existing literature [6,7], we scaled up our pre-training corpus by appending VG with COCO2017, consisting of 231k images.
> >
> > We report below our updated results on referring expression comprehension:
> >
> > Table 4: **Multi-modal fine-grained downstream: referring expression comprehension.** Best comparable results are in **bold**.
> >
> > | Method | Im-Txt | Im-Txt-Box | RefCOCO (val/testA/testB) | RefCOCO+ (val/testA/testB) | RefCOCOg (val/test) |
> > | :----: | :----: | :-----: | :--------: | :-----------------: | :-------------: |
> > | MAttNet | - | - | 76.4/80.4/69.3 | 64.9/70.3/56.0 | 66.7/67.0 |
> > | VLBERT | 3M | − | −/−/− | 71.6/77.7/61.0| −/−|
> > | ViLBERT | 3M | − | −/−/− | 72.3/78.5/62.6 | −/− |
> > | Ernie-VL-Large | 4M | − | −/−/− | 75.9/82.4/66.9| −/− |
> > | Rosita | 4M | − | 84.8/88.0/78.3 | 76.1/82.0/67.4 | 78.2/78.3 |
> > | UNITER-L | 4M | − | 81.4/87.0/74.2 | 75.9/81.5/66.7 | 74.9/75.8 |
> > | VILLA-L | 4M | − | 82.4/87.5/74.8 | 76.2/81.5/66.9 | 76.2/76.7 |
> > | MDETR-B | − | 1.3M | 87.5/90.4/82.7 | 81.1/85.5/73.0 | 83.4/83.3 |
> > | UniTAB | − | 1.3M | 86.3/88.8/80.6 | 78.7/83.2/69.5 | 80.0/80.0 |
> > | **VoLTA-B** | **231k** | **−** | **86.1/88.6/81.8** | **77.0/82.7/67.8** | **78.3/78.3** |
> >
> >
> > We can see that VoLTA beats all base-sized and larger models trained without image-text-box data across all validation and test sets of RefCOCO, RefCOCO+, and RefCOCOg. Moreover, VoLTA performs comparably with MDETR and UniTAB, even without being trained on grounding data. These results prove that our system can achieve impressive localization capability by only utilizing a global image-caption corpus.
> >
> > The same phenomenon is also seen in language-conditioned object detection tasks. We highlight that VoLTA is the first work pre-trained without image-text-box data to evaluate on these two challenging fine-grained datasets.
> >
> > Table 5: **Multi-modal fine-grained downstream: language-conditioned object detection on COCO and LVIS.** All available baselines are pre-trained on Im-Txt-Box data, while VoLTA uses Im-Txt data.
> >
> > | Method | Im-Txt-Box | COCO Val 2017 (AP) | LVIS MiniVal (APr/APc/APf/AP) |
> > | :----: | :----: | :-----: | :-----: |
> > | Mask R-CNN | &check; | − | 26.3/34.0/33.9/33.3 |
> > | MDETR | &check; | − | 20.9/24.9/24.3/24.2 |
> > | GLIP-B | &check; | 57.0 | 31.3/48.3/56.9/51.0 |
> > | **VoLTA-B** | − | **51.6** | **34.4/43.1/43.8/42.7** |
> >
> >
> > This table shows that VoLTA outperforms several strong baselines trained with grounding data on LVIS . For example, VoLTA beats Mask R-CNN, MDETR, and GLIP-B on LVIS APr, which denotes average precision on rare objects. Moreover, VoLTA performs consistently well on all unimodal and multimodal coarse-grained tasks. These results prove the success of this work’s original goal - **“eliminating the use of box annotations, but achieving superior performance on fine-grained region-level tasks without compromising on coarse-grained tasks.”**
> >
> > We added these results of larger-scale pre-training in **Tables 3, 4, and 5**.
> >
> > [6] Yen-Chun Chen et al.; UNITER: UNiversal Image-TExt Representation Learning. ECCV, 2020.
> >
> > [7] Zhicheng Huang et al.; Seeing Out of the Box: End-to-End Pre-Training for Vision-Language Representation Learning. CVPR, 2021.

---

> > > ### Author Response · Authors · 2022-11-14
> > > **General Response to All Reviewers (Part 4/4)**
> > >
> > > **Q4: What’s the advantage of using GOT compared with existing local alignment methods such as OT? Additional ablation is needed to verify the effectiveness of GOT.**
> > >
> > > We thank the reviewers (Reviewer mig1 and Reviewer PoWP) for raising this important question. We have conducted additional ablation experiments to clarify the efficacy of GOT over traditional OT alignment algorithms for different fine-grained multi-modal downstream tasks.
> > >
> > > This question is also related to Reviewer PoWP asking for evidence on how GWD can preserve graph topological information, resulting in a significantly superior alignment for intricate image-text pairs. Hence, we first recapitulate the motivation behind using GOT in our framework and then support our claims with ablation experiments.
> > >
> > > As discussed in Section 3.1.2 in the manuscript, existing VLP algorithms using OT for image-text alignment [6,8] only utilize traditional Wasserstein distance (WD). From the definition of WD in Equation 4 in the manuscript, we can see that WD is optimized for matching nodes in different graphs, failing to capture similarities between edges. Hence, WD can not distinguish between similar entities represented by other nodes in the same graph.
> > >
> > > Let us consider a simple example from the original GOT [9] paper. Given the sentence “there is a red book on the blue desk” paired with an image containing several desks and books in different colors, WD can not correctly distinguish which colored book and desk the sentence refers to. Hence, WD loses neighboring node information during alignment and fails to understand the relationships among similar objects in the image.
> > >
> > > On the other hand, Gromov-Wasserstein distance (GWD) can capture edge similarities between graphs but needs help understanding node similarity. Thus, WD and GWD provide complementary advantages during fine-grained alignment. Hence, we couple WD and GWD mutually beneficially and use a joint transport plan for accurate patch-word matching.
> > >
> > > We justify these arguments with our detailed ablation experiments.
> > >
> > > Table E.2: **Ablation study on different losses of the training objective of VoLTA for referring expression comprehension tasks.** Each model is pre-trained on 231k image-text samples from COCO2017 and VG.
> > >
> > > | $\mathcal{L}_{\mathrm{BT}}$ |  $\mathcal{L}_{\mathrm{w}}$ |  $\mathcal{L}_{\mathrm{gw}}$ |  $\mathcal{L}_{\mathrm{MLM}}$ |  $\mathcal{L}_{\mathrm{ITM}}$ | RefCOCO (val/testA/testB) | RefCOCO+ (val/testA/testB) | RefCOCOg (val/test) |
> > > | :----: | :----: | :-----: | :-----: | :----: | :----: | :-----: | :-----: |
> > > | &check; | − | − | − | − | 81.7/84.1/77.8 | 71.2/76.6/62.2 | 71.7/71.7 |
> > > | &check; | − | − | &check; | &check; | 82.7/85.2/78.1 | 72.0/77.7/62.5 | 72.8/72.7 |
> > > | &check; | &check; | − | &check; | &check; | 83.9/86.6/80.5 | 73.9/79.5/64.1 | 74.6/74.3 |
> > > | &check; | &check; | &check; | &check; | &check; | **86.1/88.6/81.8** | **77.0/82.7/67.8** | **78.3/78.3** |
> > >
> > >
> > > As shown in Table E.2, we analyze the effectiveness of different pre-training objectives through an ablation on fine-grained downstream tasks. First, we pre-train VoLTA only with the multi-modal BT loss. Next, we add MLM and ITM loss which helps the model to learn cross-modal information via attention fusion. Next, we add the GOT pre-training objective. Note that GOT adopts two types of OT distances - WD for node matching and GWD for edge matching. As shown in Table E.2, the GWD loss helps improve reference expression comprehension performance across RefCOCO, RefCOCO+, and RefCOCOg datasets. Specifically on RefCOCOg, adding the GWD loss yields a significant 4.0\% boost in the challenging test set. This dataset contains intricate images with multiple similar objects with different shapes and colors; hence GWD is crucial in distinguishing between them. Overall, this set of experiments demonstrates that all of the objectives are necessary for our model to obtain good performance on different fine-grained multi-modal tasks.
> > >
> > > We added this result in the **Appendix Table E.2.**
> > >
> > > [6] Yen-Chun Chen et al.; UNITER: UNiversal Image-TExt Representation Learning. ECCV, 2020.
> > >
> > > [8] Wonjae Kim et al.; ViLT: Vision-and-language transformer without convolution or region supervision. ICML, 2021.
> > >
> > > [9] Liqun Chen et al.; Graph optimal transport for cross-domain alignment. ICML, 2020.

---

### Author Response · Authors · 2022-11-25
**Note to Area Chairs, Senior Area Chairs, and Program Chairs: We are Waiting for the Reviewers' Feedback**

Dear Area Chairs, Senior Area Chairs, and Program Chairs,

In our rebuttal, we have addressed all of the reviewers' concerns in depth. However, we are still waiting for the responses from the reviewers. We want to thank the reviewers for their comments, which helped us significantly improve the clarity and increase the scope of the paper. We addressed the key concerns by

1. Clarifying the motivation behind our work and explaining how this work differs from the wide range of existing vision-language pre-training (VLP) literature.
2. Clarifying our training strategy and reiterating the benefits of using gated-attention over traditional cross-attention transformer layers.
3. Additional larger-scale pre-training experiments and additional ablation studies verifying the advantages of using GOT compared with existing alignment methods such as OT.
4. Modifying Figure 2 to indicate the connections between three submodules.

We also want to highlight that some of the comments raised by **Reviewer PoWP** lack proper comprehension of the motivation and basic details of our proposed method. Hence, such a review fails to perform a holistic evaluation of the significance of our work on generalized, unified, and scalable vision-language pre-training. We provide some examples here:

1. Reviewer PoWP asked, $\textcolor{brown}{\text{``What}}$ $\textcolor{brown}{\text{if}}$ $\textcolor{brown}{\text{contrastive-based}}$ $\textcolor{brown}{\text{methods}}$ $\textcolor{brown}{\text{use the}}$ $\textcolor{brown}{\text{same}}$ $\textcolor{brown}{\text{batch}}$ $\textcolor{brown}{\text{size as}}$ $\textcolor{brown}{\text{BT?}}$ $\textcolor{brown}{\text{Will}}$ $\textcolor{brown}{\text{it}}$ $\textcolor{brown}{\text{degrade}}$  $\textcolor{brown}{\text{the}}$ $\textcolor{brown}{\text{performance}}$ $\textcolor{brown}{\text{significantly?}}$ $\textcolor{brown}{\text{If}}$ $\textcolor{brown}{\text{not,}}$ $\textcolor{brown}{\text{then}}$ $\textcolor{brown}{\text{we}}$ $\textcolor{brown}{\text{can}}$ $\textcolor{brown}{\text{use}}$ $\textcolor{brown}{\text{small}}$ $\textcolor{brown}{\text{batch}}$ $\textcolor{brown}{\text{sizes."}}$ However, It is an established and widely-discussed fact in the literature that contrastive objectives suffer substantially with small batch sizes. We have explained this concept in detail in "Response to Reviewer PoWP" Q1.

2. Reviewer PoWP asked, $\textcolor{brown}{\text{``On}}$ $\textcolor{brown}{\text{page}}$ $\textcolor{brown}{\text{5:}}$ $\textcolor{brown}{\text{WD}}$ $\textcolor{brown}{\text{can}}$ $\textcolor{brown}{\text{only}}$ $\textcolor{brown}{\text{match}}$ $\textcolor{brown}{\text{nodes}}$ $\textcolor{brown}{\text{in}}$ $\textcolor{brown}{\text{the}}$ $\textcolor{brown}{\text{graph,}}$ $\textcolor{brown}{\text{and}}$ $\textcolor{brown}{\text{will}}$ $\textcolor{brown}{\text{treat}}$ $\textcolor{brown}{\text{all}}$ $\textcolor{brown}{\textit{men}}$ $\textcolor{brown}{\text{entities}}$ $\textcolor{brown}{\text{ as}}$ $\textcolor{brown}{\text{identical}}$ $\textcolor{brown}{\text{and}}$ $\textcolor{brown}{\text{will}}$ $\textcolor{brown}{\text{ignore}}$ $\textcolor{brown}{\text{neighboring}}$ $\textcolor{brown}{\text{relations}}$ $\textcolor{brown}{\text{like}}$ $\textcolor{brown}{\textit{in}}$ $\textcolor{brown}{\textit{blue}}$ $\textcolor{brown}{\textit{shirt}}$ $\textcolor{brown}{\text{and}}$ $\textcolor{brown}{\textit{holding}}$ $\textcolor{brown}{\textit{the}}$ $\textcolor{brown}{\textit{scissors.}}$ $\textcolor{brown}{\text{Is there}}$ $\textcolor{brown}{\text{any}}$ $\textcolor{brown}{\text{evidence to}}$ $\textcolor{brown}{\text{support this}}$ $\textcolor{brown}{\text{statement?"}}$ We have provided detailed reasons why WD can not distinguish between similar entities represented by other nodes in the same graph in section 3.1.2 of our manuscript. We have also clarified this fact again in "Response to All Reviewers" Q4.

3. Reviewer PoWP asked, $\textcolor{brown}{\text{``How was GOT}}$ $\textcolor{brown}{\text{applied}}$ $\textcolor{brown}{\text{to image}}$ $\textcolor{brown}{\text{and text}}$ $\textcolor{brown}{\text{features?}}$ $\textcolor{brown}{\text{This is not}}$ $\textcolor{brown}{\text{discussed}}$ $\textcolor{brown}{\text{in the}}$ $\textcolor{brown}{\text{Method}}$ $\textcolor{brown}{\text{section."}}$ However, we thoroughly explained this alignment strategy in Method section 3.1.2 and our pseudo code (Appendix A).


We understand that reviewing can be difficult in strict and crisp timelines; hence, some details may remain overlooked during the initial review. This is precisely the significance of ICLR's interactive rebuttal, which helps both the reviewers and the authors understand the work, get feedback, and improve the manuscript. However, a one-sided conversation defeats the whole purpose. We would be grateful if the reviewers could confirm whether our response has addressed their concerns and let us know if any issues remain.

Thanks, and Best Regards, \
The authors

---

### Decision · Program_Chairs · 2023-01-20

**Decision:**

Reject

**Justification For Why Not Higher Score:**

No reviewer argues for acceptance (except for a "marginally above threshold").

**Justification For Why Not Lower Score:**

N/A

**Metareview: Summary, Strengths And Weaknesses:**

The paper proposes graph optimal transport for aligning patches and words in vision-language pretraining. The effect is demonstrated on visual recognition and VQA datasets. While the reviewers appreciate some of the ideas and experiments (e.g. Barlow Twins for image-text), they question the motivation, find several details unclear, and some parts of the results not significant.